# State-Action Similarity-Based Representations for Off-Policy Evaluation

**Brahma S. Pavse** and **Josiah P. Hanna**
University of Wisconsin – Madison
`pavse@wisc.edu, jphanna@cs.wisc.edu`

## Abstract

In reinforcement learning, off-policy evaluation (OPE) is the problem of estimating the expected return of an evaluation policy given a fixed dataset that was collected by running one or more different policies. One of the more empirically successful algorithms for OPE has been the fitted q-evaluation (FQE) algorithm that uses temporal difference updates to learn an action-value function, which is then used to estimate the expected return of the evaluation policy. Typically, the original fixed dataset is fed directly into FQE to learn the action-value function of the evaluation policy. Instead, in this paper, we seek to enhance the data-efficiency of FQE by first transforming the fixed dataset using a learned encoder, and then feeding the transformed dataset into FQE. To learn such an encoder, we introduce an OPE-tailored state-action behavioral similarity metric, and use this metric and the fixed dataset to learn an encoder that models this metric. Theoretically, we show that this metric allows us to bound the error in the resulting OPE estimate. Empirically, we show that other state-action similarity metrics lead to representations that cannot represent the action-value function of the evaluation policy, and that our state-action representation method boosts the data-efficiency of FQE and lowers OPE error relative to other OPE-based representation learning methods on challenging OPE tasks. We also empirically show that the learned representations significantly mitigate divergence of FQE under varying distribution shifts. Our code is available here: `https://github.com/Badger-RL/ROPE`.

## 1 Introduction

In real life applications of reinforcement learning, practitioners often wish to assess the performance of a learned policy before allowing it to make decisions with real life consequences [Theocharous et al., 2015]. That is, they want to be able to evaluate the performance of a policy without actually deploying it. One approach of accomplishing this goal is to apply methods for off-policy evaluation (OPE). OPE methods evaluate the performance of a given evaluation policy using a fixed offline dataset previously collected by one or more policies that may be different from the evaluation policy.

One of the core challenges in OPE is that the offline datasets may have limited size. In this situation, it is often critical that OPE algorithms are data-efficient. That is, they are able produce accurate estimates of the evaluation policy value even when only small amounts of data are available. In this paper, we seek to enhance the data-efficiency of OPE methods through representation learning. While prior works have studied representation learning for OPE, they have mostly considered representations that induce guaranteed convergent learning without considering whether data-efficiency increases [Chang et al., 2022, Wang et al., 2021]. For example, Chang et al. [2022] introduce a method for learning Bellman complete representations for FQE but empirically find that having such a learned representation provides little benefit compared to FQE without the learned representation. Thus, in this work we ask the question, "can explicit representation learning lead to more data-efficient OPE?"

37th Conference on Neural Information Processing Systems (NeurIPS 2023).

To answer this question, we take inspiration from recent advances in learning state similarity metrics for control [Castro et al., 2022, Zhang et al., 2021a]. These works define behavioral similarity metrics that measure the distance between two states. They then show that state representations can be learned such that states that are close under the metric will also have similar representations. In our work, we introduce a new OPE-tailored behavioral similarity metric called **R**epresentations for **O**ff-**P**olicy **E**valuation (ROPE) and show that learning ROPE representations can lead to more accurate OPE.

Specifically, ROPE first uses the fixed offline dataset to learn a state-action encoder based on this OPE-specific state-action similarity metric, and then applies this encoder to the same dataset to produce a new representation for all state-action pairs. The transformed data is then fed into the fitted q-evaluation (FQE) algorithm [Le et al., 2019] to produce an OPE estimate. We theoretically show that the error between the policy value estimate with FQE + ROPE and the true evaluation policy value is upper-bounded in terms of how ROPE aggregates state-action pairs. We empirically show that ROPE improves the data-efficiency of FQE and leads to lower OPE error compared to other OPE-based representation learning baselines. Additionally, we empirically show that ROPE representations mitigate divergence of FQE under extreme distribution. To the best of our knowledge, our work is the first to propose an OPE-specific state-action similarity metric that increases the data-efficiency of OPE.

## 2 Background

In this section, we formalize our problem setting and discuss prior work.

### 2.1 Notation and Problem Setup

We consider an infinite-horizon Markov decision process (MDP) [Puterman, 2014], $\mathcal{M} = \langle \mathcal{S}, \mathcal{A}, \mathcal{R}, P, \gamma, d_0 \rangle$, where $\mathcal{S}$ is the state-space, $\mathcal{A}$ is the action-space, $\mathcal{R} : \mathcal{S} \times \mathcal{A} \rightarrow \Delta([0, \infty))$ is the reward function, $P : \mathcal{S} \times \mathcal{A} \rightarrow \Delta(\mathcal{S})$ is the transition dynamics function, $\gamma \in [0, 1)$ is the discount factor, and $d_0 \in \Delta(\mathcal{S})$ is the initial state distribution, where $\Delta(X)$ is the set of all probability distributions over a set $X$. We refer to the joint state-action space as $\mathcal{X} := \mathcal{S} \times \mathcal{A}$. The agent acting, according to policy $\pi$, in the MDP generates a trajectory: $S_0, A_0, R_0, S_1, A_1, R_1, ...$, where $S_0 \sim d_0$, $A_t \sim \pi(\cdot|S_t)$, $R_t \sim \mathcal{R}(S_t, A_t)$, and $S_{t+1} \sim P(\cdot|S_t, A_t)$ for $t \geq 0$. We define $r(s, a) := \mathbb{E}[\mathcal{R}(s, a)]$.

We define the performance of policy $\pi$ to be its expected discounted return, $\rho(\pi) := \mathbb{E}[\sum_{t=0}^{\infty} \gamma^t R_t]$. We then have the action-value function of a policy for a given state-action pair, $q^\pi(s, a) = r(s, a) + \gamma \mathbb{E}_{S' \sim P(s,a), A' \sim \pi}[q^\pi(S', A')]$, which gives the expected discounted return when starting in state $s$ and then taking action $a$. Then $\rho(\pi)$ can also be expressed as $\rho(\pi) = \mathbb{E}_{S_0 \sim d_0, A_0 \sim \pi}[q^\pi(S_0, A_0)]$.

It is often more convenient to work with vectors instead of atomic states and actions. We use $\phi : \mathcal{S} \times \mathcal{A} \rightarrow \mathbb{R}^d$ to denote a representation function that maps state-action pairs to vectors with some dimensionality $d$.

### 2.2 Off-Policy Evaluation (OPE)

In off-policy evaluation, we are given a fixed dataset of $m$ transition tuples $\mathcal{D} := \{(s_i, a_i, s_i', r_i)\}_{i=1}^m$ and an evaluation policy, $\pi_e$. Our goal is to use $\mathcal{D}$ to estimate $\rho(\pi_e)$. Crucially, $\mathcal{D}$ may have been generated by a set of *behavior* policies that are different from $\pi_e$, which means that simply averaging the discounted returns in $\mathcal{D}$ will produce an inconsistent estimate of $\rho(\pi_e)$. We do *not* assume that these behavior policies are known to us, however, we do make the standard assumption that $\forall s \in \mathcal{S}, \forall a \in \mathcal{A}$ if $\pi_e(a|s) > 0$ then the state-action pair $(s, a)$ has non-zero probability of appearing in $\mathcal{D}$.

As done by Fu et al. [2021], we measure the accuracy of an OPE estimator with the *mean absolute error* (MAE) to be robust to outliers. Let $\hat{\rho}(\pi_e, \mathcal{D})$ be the estimate returned by an OPE method using $\mathcal{D}$. The MAE of this estimate is given as:

$$\text{MAE}[\hat{\rho}] := \mathbb{E}_{\mathcal{D}}[|\hat{\rho}(\pi_e, \mathcal{D}) - \rho(\pi_e)|].$$

While in practice $\rho(\pi_e)$ is unknown, it is standard for the sake of empirical analysis [Voloshin et al., 2021, Fu et al., 2021] to estimate it by executing rollouts of $\pi_e$.

## 2.3 Fitted Q-Evaluation

One of the more successful OPE methods has been fitted q-evaluation (FQE) which uses batch temporal difference learning [Sutton, 1988] to estimate $\rho(\pi_e)$ [Le et al., 2019]. FQE involves two conceptual steps: 1) repeat temporal difference policy evaluation updates to estimate $q^{\pi_e}(s, a)$ and then 2) estimate $\rho(\pi_e)$ as the mean action-value at the initial state distribution. Formally, let the action-value function be parameterized by $\xi$ i.e. $q_\xi$, then the following loss function is minimized to estimate $q^{\pi_e}$:

$$\mathcal{L}_{\text{FQE}}(\xi) := \mathbb{E}_{(s,a,s',r)\sim\mathcal{D}}\left[\left(r(s,a) + \gamma\,\mathbb{E}_{a'\sim\pi_e(\cdot|s')}[q_{\bar{\xi}}(s',a')] - q_\xi(s,a)\right)^2\right]$$

where $\bar{\xi}$ is a separate copy of the parameters $\xi$ and acts as the target function approximator [Mnih et al., 2015] that is updated to $\xi$ at a certain frequency. The learned $q_{\xi^*}$ is then used to estimate the policy value: $\hat{\rho}(\pi_e) := \mathbb{E}_{s_0\sim d_0, a_0\sim\pi_e}[q_{\xi^*}(s_0, a_0)]$. While conceptually FQE can be implemented with many classes of function approximator to represent the $q_\xi$, in practice, deep neural networks are often the function approximator of choice. When using deep neural networks, FQE can be considered a policy evaluation variant of neural fitted q-iteration [Riedmiller, 2005].

## 2.4 Related Work

In this section, we discuss the most relevant prior literature on off-policy evaluation and representation learning. Methods for OPE are generally categorized as importance-sampling based [Precup et al., Thomas et al., Hanna et al., 2021, Liu et al., 2018, Yang et al., 2020b], model-based [Yang and Nachum, 2021, Zhang et al., 2021b, Hanna et al., 2017], value-function-based [Le et al., 2019, Uehara et al., 2020], or hybrid [Jiang and Li, 2016, Thomas and Brunskill, 2016, Farajtabar et al., 2018]. Our work focuses on FQE, which is a representative value-function-based method, since it has been shown to have strong empirical performance [Fu et al., 2021, Chang et al., 2022]. We refer the reader to Levine et al. [2020] for an in-depth survey of OPE methods.

**Representation Learning for Off-policy Evaluation and Offline RL**  A handful of works have considered the interplay of representation learning with OPE methods and offline RL. Yang and Nachum [2021] benchmark a number of existing representation learning methods for offline RL and show that pre-training representation can be beneficial for offline RL. They also consider representation learning based on behavioral similarity and find that such representations do not enable successful offline RL. However, their study is focused on evaluating existing algorithms and on control. Pavse and Hanna [2023] introduced state abstraction [Li et al., 2006] as an approach to lower the variance of OPE estimates in importance-sampling based methods. However, their work made the strict assumption of granting access to a bisimulation abstraction in theory and relied on a hand-specified abstraction in practice. Only recently have works started to consider learning representations specifically for OPE. Chang et al. [2022] introduced a method for learning Bellman complete representations that enabled convergent approximation of $q^{\pi_e}$ with linear function approximation. Wang et al. [2021] show that using the output of the penultimate layer of $\pi_e$'s action-value function provides realizability of $q_{\pi_e}$, but is insufficient for accurate policy evaluation under extreme distribution shift. Our work explicitly focuses on boosting the data-efficiency of OPE methods and lowers the error of OPE estimates compared to Chang et al. [2022] and Wang et al. [2021].

**Representation Learning via Behavioral Similarity**  The representation learning method we introduce builds upon prior work in learning representations in which similar states share similar representations. Much of this prior work is based on the notion of a bisimulation abstraction in which two states with identical reward functions and that lead to identical groups of next states should be classified as similar [Ferns et al., 2004, 2011, Ferns and Precup, 2014, Castro, 2019]. The bisimulation metric itself is difficult to learn both computationally and statistically and so recent work has introduced various approximations [Castro et al., 2022, Castro, 2019, Zhang et al., 2021a, Gelada et al., 2019]. To the best of our knowledge, all of this work has considered the *online, control* setting and has only focused on state representation learning. In contrast, we introduce a method for learning *state-action* representations for OPE with a fixed dataset. One exception is the work of Dadashi et al. [2021], which proposes to learn state-action representations for offline policy *improvement*. However, as we will show in Section 4, the distance metric that they base their representations on is inappropriate in the OPE context.

# 3 ROPE: State-Action Behavioral Similarity Metric for Off-Policy Evaluation

In this section, we introduce our primary algorithm: **R**epresentations for OPE (ROPE), a representation learning method based on state-action behavioral similarity that is tailored to the off-policy evaluation problem. That is, using a fixed off-policy dataset $\mathcal{D}$, ROPE learns similar representations for state-action pairs that are similar in terms of the action-value function of $\pi_\mathrm{e}$.

Prior works on representation learning based on state behavioral similarity define a metric that relates the similarity of two states and then map similar states to similar representations [Castro et al., 2022, Zhang et al., 2021a]. We follow the same high-level approach except we focus instead on learning state-action representations for OPE. One advantage of learning state-action representations over state representations is that we can learn a metric specifically for $\pi_\mathrm{e}$ by directly sampling actions from $\pi_\mathrm{e}$ instead of using importance sampling, which can be difficult when the multiple behavior policies are unknown. Moreover, estimating the importance sampling ratio from data is known to be challenging [Hanna et al., 2021, Yang et al., 2020a].

Our new notion of similarity between state-action pairs is given by the recursively-defined ROPE distance, $d_{\pi_\mathrm{e}}(s_1, a_1; s_2, a_2) := |r(s_1, a_1) - r(s_2, a_2)| + \gamma\, \mathbb{E}_{s'_1, s'_2 \sim P, a'_1, a'_2 \sim \pi_e}[d_{\pi_\mathrm{e}}(s'_1, a'_1; s'_2, a'_2)]$. Intuitively, $d_{\pi_\mathrm{e}}$ measures how much two state-action pairs, $(s_1, a_1)$ and $(s_2, a_2)$, differ in terms of short-term reward and discounted expected distance between next state-action pairs encountered by $\pi_\mathrm{e}$. In order to compute $d_{\pi_\mathrm{e}}$, we define the ROPE operator:

**Definition 1** (ROPE operator). *Given an evaluation policy $\pi_\mathrm{e}$, the* ROPE *operator $\mathcal{F}^{\pi_e} : \mathbb{R}^{\mathcal{X} \times \mathcal{X}} \to \mathbb{R}^{\mathcal{X} \times \mathcal{X}}$ is given by:*

$$\mathcal{F}^{\pi_e}(d)(s_1, a_1; s_2, a_2) := \underbrace{|r(s_1, a_1) - r(s_2, a_2)|}_{\text{short-term distance}} + \gamma \underbrace{\mathbb{E}_{s'_1, s'_2 \sim P, a'_1, a'_2 \sim \pi_e}[d(s'_1, a'_1; s'_2, a'_2)]}_{\text{long-term distance}} \quad (1)$$

*where $d : \mathcal{X} \times \mathcal{X} \to \mathbb{R}$, $s'_1 \sim P(s'_1|s_1, a_1), s'_2 \sim P(s'_2|s_2, a_2), a'_1 \sim \pi_\mathrm{e}(\cdot|s'_1), a'_2 \sim \pi_\mathrm{e}(\cdot|s'_2)$*

Given the operator, $\mathcal{F}^{\pi_e}$, we show that the operator is a contraction mapping, computes the ROPE distance, $d_{\pi_\mathrm{e}}$, and that $d_{\pi_\mathrm{e}}$ is a *diffuse metric*. For the background on metrics and full proofs, refer to the Appendix A and B.

**Proposition 1.** *The operator $\mathcal{F}^{\pi_e}$ is a contraction mapping on $\mathbb{R}^{\mathcal{X} \times \mathcal{X}}$ with respect to the $L^\infty$ norm.*

**Proposition 2.** *The operator $\mathcal{F}^{\pi_e}$ has a unique fixed point $d_{\pi_e} \in \mathbb{R}^{\mathcal{X} \times \mathcal{X}}$. Let $d_0 \in \mathbb{R}^{\mathcal{X} \times \mathcal{X}}$, then $\lim_{t \to \infty} \mathcal{F}_t^{\pi_e}(d_0) = d_{\pi_e}$.*

Propositions 1 and 2 ensure that repeatedly applying the operator on some function $d : \mathcal{X} \times \mathcal{X} \to \mathbb{R}$ will make $d$ converge to our desired distance metric, $d_{\pi_\mathrm{e}}$. An important aspect of $d_{\pi_\mathrm{e}}$ is that it is a diffuse metric:

**Proposition 3.** *$d_{\pi_\mathrm{e}}$ is a diffuse metric.*

where a diffuse metric is the same as a psuedo metric (see Definition 3 in Appendix A) except that self-distances can be non-zero i.e. it may be true that $d_{\pi_\mathrm{e}}(s, a; s, a) > 0$. This fact arises due to the stochasticity in the transition dynamics and action sampling from $\pi_\mathrm{e}$. If we assume a deterministic transition function and a deterministic $\pi_e$, $d_{\pi_e}$ will reduce to a pseudo metric, which gives zero self-distance. In practice, we use a sample approximation of the ROPE operator to estimate $d_{\pi_\mathrm{e}}$.

Given that $d_{\pi_\mathrm{e}}$ is well-defined, we have the following theorem that shows why it is useful in the OPE context:

**Theorem 1.** *For any evaluation policy $\pi_e$ and $(s_1, a_1), (s_2, a_2) \in \mathcal{X}$, we have that $|q^{\pi_e}(s_1, a_1) - q^{\pi_e}(s_2, a_2)| \leq d_{\pi_e}(s_1, a_1, ; s_2, a_2)$.*

Given that our goal is learn representations based on $d_{\pi_\mathrm{e}}$, Theorem 1 implies that whenever $d_{\pi_\mathrm{e}}$ considers two state-action pairs to be close or have similar representations, they will also have close action-values. In the context of OPE, if the distance metric considers two state-action pairs that have *different* action-values to be zero distance apart/have the same representation, then FQE will have to output two different action-values for the same input representation, which inevitably means FQE must be inaccurate for at least one state-action pair.

## 3.1 Learning State-Action Representations with ROPE

In practice, our goal is to use $d_{\pi_e}$ to learn a state-action representation $\phi(s, a) \in \mathbb{R}^d$ such that the distances between these representations matches the distance defined by $d_{\pi_e}$. To do so, we follow the approach by Castro et al. [2022] and directly parameterize the value $d_{\pi_e}(s_1, a_1; s_2, a_2)$ as follows:

$$d_{\pi_e}(s_1, a_1; s_2, a_2) \approx \tilde{d}_\omega(s_1, a_1; s_2, a_2) := \frac{||\phi_\omega(s_1, a_1)||_2^2 + ||\phi_\omega(s_2, a_2)||_2^2}{2}$$
$$+ \beta\theta(\phi_\omega(s_1, a_1), \phi_\omega(s_2, a_2)) \quad (2)$$

in which $\phi$ is parameterized by some function approximator whose parameter weights are denoted by $\omega$, $\theta(\cdot, \cdot)$ gives the angular distance between the vector arguments, and $\beta$ is a parameter controlling the weight of the angular distance. We can then learn the desired $\phi_\omega$ through a sampling-based bootstrapping procedure [Castro et al., 2022]. More specifically, the following loss function is minimized to learn the optimal $\omega^*$:

$$\mathcal{L}_{\text{ROPE}}(\omega) := \mathbb{E}_\mathcal{D}\left[\left(\left(|r(s_1, a_1) - r(s_2, a_2)| + \gamma\,\mathbb{E}_{\pi_e}[\tilde{d}_{\bar{\omega}}(s_1', a_1'; s_2', a_2')] - \tilde{d}_\omega(s_1, a_1; s_2, a_2)\right)^2\right]$$
$$(3)$$

where $\bar{\omega}$ is separate copy of $\omega$ and acts as a target function approximator [Mnih et al., 2015], which is updated to $\omega$ at a certain frequency. Once $\phi_{\omega^*}$ is obtained using $\mathcal{D}$, we use $\phi_{\omega^*}$ with FQE to perform OPE with the same data. Conceptually, the FQE procedure is unchanged except the learned action-value function now takes $\phi_{\omega^*}(s, a)$ as its argument instead of the state and action directly.

With ROPE, state-action pairs are grouped together when they have small pairwise ROPE distance. Thus, a given group of state-action pairs have similar state-action representations and are behaviorally similar (i.e, have similar rewards and lead to similar future states when following $\pi_e$). Consequently, these state-action pairs will have a similar action-value, which allows data samples from any member of the group to learn the group's shared action-value as opposed to learning the action-value for each state-action pair individually. This generalized usage of data leads to more data-efficient learning. We refer the reader to Appendix C for ROPE's pseudo-code.

## 3.2 Action-Value and Policy Value Bounds

We now theoretically analyze how ROPE state-action representations help FQE estimate $\rho(\pi_e)$. For this analysis, we focus on hard groupings where groups of similar state-action pairs are aggregated into one cluster and no generalization is performed across clusters; in practice, we learn state-action representations in which the difference between representations approximates the ROPE distance between state-action pairs. Furthermore, for theoretical analysis, we consider exact computation of the ROPE diffuse metric and of action-values using dynamic programming. First, we present the following lemma. For proofs, refer to Appendix B.

**Lemma 1.** *Assume the rewards* $\mathcal{R} : \mathcal{S} \times \mathcal{A} \to \Delta([0, 1])$ *then given an aggregated* MDP $\widetilde{\mathcal{M}} = \langle \widetilde{\mathcal{S}}, \widetilde{\mathcal{A}}, \widetilde{\mathcal{R}}, \widetilde{P}, \gamma, \tilde{d}_0 \rangle$ *constructed by aggregating state-actions in an $\epsilon$-neighborhood based on* $d_{\pi_e}$, *and an encoder* $\phi : \mathcal{X} \to \widetilde{\mathcal{X}}$ *that maps state-actions in $\mathcal{X}$ to these clusters, the action-value for the evaluation policy $\pi_e$ in the two* MDPs *are bounded as:*

$$|q^{\pi_e}(x) - \tilde{q}^{\pi_e}(\phi(x))| \leq \frac{2\epsilon}{(1 - \gamma)}$$

Lemma 1 states that the error in our estimate of the true action-value function of $\pi_e$ is upper-bounded by the clustering radius of $d_{\pi_e}$, $\epsilon$. Lemma 1 then leads us to our main result:

**Theorem 2.** *Under the same conditions as Lemma 1, the difference between the expected fitted q-evaluation (*FQE*) estimate and the expected estimate of* FQE+ROPE *is bounded:*

$$\left| \mathbb{E}_{s_0, a_0 \sim \pi_e}[q^{\pi_e}(s_0, a_0)] - \mathbb{E}_{s_0, a_0 \sim \pi_e}[q^{\pi_e}(\phi(s_0, a_0))] \right| \leq \frac{2\epsilon}{(1 - \gamma)}$$

Theorem 2 tells us that the error in our estimate of $\rho(\pi_e)$ is upper-bounded by the size of the clustering radius $\epsilon$. The implication is that grouping state-action pairs according to the ROPE diffuse metric enables us to upper bound error in the OPE estimate. At an extreme, if we only group state-action pairs with *zero* ROPE distance together then we obtain zero absolute error meaning that the action-value function for the aggregated MDP is able to realize the action-value function of the original MDP.

# 4 Empirical Study

In this section, we present an empirical study of ROPE designed to answer the following questions:

1. Does ROPE group state-actions that are behaviorally similar according to $q^{\pi_e}$?
2. Does ROPE improve the data-efficiency of FQE and achieve lower OPE error than other OPE-based representation methods?
3. How sensitive is ROPE to hyperparameter tuning and extreme distribution shifts?

## 4.1 Empirical Set-up

We now describe the environments and datasets used in our experiments.

**Didactic Domain.** We provide intuition about ROPE on our gridworld domain. In this tabular and deterministic environment, an agent starts from the bottom left of a $3 \times 3$ grid and moves to the terminal state at the top right. The reward function is the negative of the Manhattan distance from the top right. $\pi_e$ stochastically moves up or right from the start state and then deterministically moves towards the top right, and moves deterministically right when it is in the center. The behavior policy $\pi_b$ acts uniformly at random in each state. We set $\gamma = 0.99$.

**High-Dimensional Domains.** We conduct our experiments on five domains: HumanoidStandup, Swimmer, HalfCheetah, Hopper, and Walker2D, each of which has 393, 59, 23, 14, and 23 as the native state-action dimension respectively. We set $\gamma = 0.99$.

**Datasets.** We consider 12 different datasets: 3 custom datasets for HumanoidStandup, Swimmer, and HalfCheetah; and 9 D4RL datasets [Fu et al., 2020] for HalfCheetah, Hopper, and Walker2D. Each of the three custom datasets is of size 100K transition tuples with an equal split between samples generated by $\pi_e$ and a lower performing behavior policy. For the D4RL datasets, we consider three types for each domain: random, medium, medium-expert, which consists of samples from a random policy, a lower performing policy, and an equal split between a lower performing and expert evaluation policy ($\pi_e$). Each dataset has 1M transition tuples. Note that due to known discrepancies between environment versions and state-action normalization procedures [1], we generate our own datasets using the publicly available policies[2] instead of using the publicly available datasets. See Appendix D for the details on the data generation procedure.

**Evaluation Protocol.** Following Fu et al. [2021], Voloshin et al. [2021] and to make error magnitudes more comparable across domains, we use relative mean absolute error (RMAE). RMAE is computed using a single dataset $\mathcal{D}$ and by generating $n$ seeds: $\mathrm{RMAE}_i(\hat{\rho}(\pi_e)) := \frac{|\rho(\pi_e) - \hat{\rho}_i(\pi_e)|}{|\rho(\pi_e) - \rho(\pi_{\mathrm{rand}})|}$, where $\hat{\rho}_i(\pi_e)$ is computed using the $i^{\mathrm{th}}$ seed and $\rho(\pi_{\mathrm{rand}})$ is the value of a random policy. We then report the Interquartile Mean (IQM) [Agarwal et al., 2021b] of these $n$ RMAEs.

**Representation learning + OPE.** Each algorithm is given access to the same fixed dataset to learn $q^{\pi_e}$. The representation learning algorithms (ROPE and baselines) use this dataset to first pre-train a representation encoder, which is then used to transform the fixed dataset. This transformed dataset is then used to estimate $q^{\pi_e}$. Vanilla FQE directly operates on the original state-action pairs.

## 4.2 Empirical Results

We now present our main empirical results.

### 4.2.1 Designing ROPE: A State-Action Behavioral Similarity Metric for OPE

The primary consideration when designing a behavioral similarity distance function for OPE, and specifically, for FQE is that the distance function should not consider two state-action pairs with different $q^{\pi_e}$ values to be the same. Suppose we have a distance function $d$, two state-actions pairs, $(s_1, a_1)$ and $(s_2, a_2)$, and their corresponding $q^{\pi_e}$. Then if $d(s_1, a_1; s_2, a_2) = 0$, it should be the case that $q^{\pi_e}(s_1, a_1) = q^{\pi_e}(s_2, a_2)$. On the other hand, if $d(s_1, a_1; s_2, a_2) = 0$ but $q^{\pi_e}(s_1, a_1)$ and $q^{\pi_e}(s_2, a_2)$ are very different, then FQE will have to output *different* action-values for the *same* input, thus inevitably making FQE inaccurate on these state-action pairs.

---

[1] https://github.com/Farama-Foundation/D4RL/tree/master
[2] https://github.com/google-research/deep_ope

While there have been a variety of proposed behavioral similarity metrics for control, they do not always satisfy the above criterion for OPE. We consider various state-action behavioral similarity metrics. Due to space constraints, we show results only for: on-policy MICO [Castro et al., 2022] $d_{\pi_b}(s_1, a_1; s_2, a_2) := |r(s_1, a_1) - r(s_2, a_2)| + \gamma \mathbb{E}_{a_1', a_2' \sim \pi_b}[d_{\pi_b}((s_1', a_1'), (s_2', a_2'))]$, which groups state-actions that have equal $q^{\pi_b}$, and defer results for the random-policy metric [Dadashi et al., 2021] and policy similarity metric [Agarwal et al., 2021a] to the Appendix D.

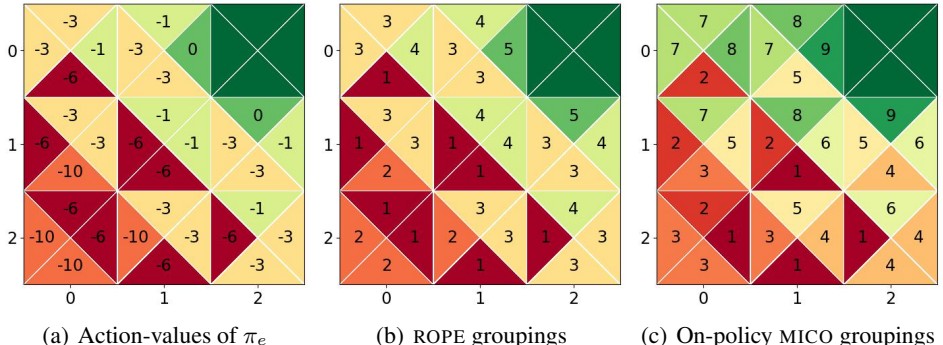

(a) Action-values of $\pi_e$      (b) ROPE groupings      (c) On-policy MICO groupings

Figure 1: Figure (a): $q^{\pi_e}$; center number in each triangle is the $q^{\pi_e}$ for that state-action pair. Center and right: group clustering according to ROPE (ours; Figure (b)) and on-policy MICO (Figure (c)) (center number in each triangle is group ID). Two state-action pairs are grouped together if their distance according to the specific metric is 0. The top right cell is blank since it is the terminal state and is not grouped.

We visualize how these different metrics group state-action pairs in our gridworld example where a state-action is represented by a triangle in the grid (Figure 1). The gridworld is $3 \times 3$ grid represented by 9 squares (states), each having 4 triangles (actions). A numeric entry in a given triangle represents either: 1) the action-value of that state-action pair for $\pi_e$ (Figure 1(a)) or 2) the group ID of the given state-action pair (Figures 1(b) and 1(c)). Along with the group ID, each state-action pair is color-coded indicating its group. In this tabular domain, we compute the distances using dynamic programming with expected updates.

The main question we answer is: does a metric group two state-action pairs together when they have the same action-values under $\pi_e$? In Figure 1(a) we see the $q^{\pi_e}$ values for each state-action where all state-action pairs that have the same action-value are grouped together under the same color (e.g. all state-action pairs with $q^{\pi_e}(\cdot, \cdot) = -6$ belong to the same group (red)). In Figure 1(b), we see that ROPE's grouping is exactly aligned with the grouping in Figure 1(a) i.e. state-action pairs that have the same action-values have the same group ID and color. On the other hand, from Figure 1(c), we see that on-policy MICO misaligns with Figure 1(a). In Appendix D, we also see similar misaligned groupings using the random-policy metric Dadashi et al. [2021] and policy similarity metric Agarwal et al. [2021a]. The misalignment of these metrics is due to the fact that they do not group state-action pairs togethers that share $q^{\pi_e}$ values.

### 4.2.2 Deep OPE Experiments

We now consider OPE in challenging, high dimensional continuous state and action space domains. We compare the RMAE achieved by an OPE algorithm using *different* state-action representations as input. If algorithm A achieves lower error than algorithm B, then A is more data-efficient than B.

**Custom Dataset Results** For the custom datasets, we consider mild distribution shift scenarios, which are typically easy for OPE algorithms. In Figure 2, we report the RMAE vs. training iterations of FQE with different state-action features fed into FQE. We consider three different state-action features: 1) ROPE (ours), 2) $\pi_e$-critic, which is a representation outputted by the penultimate layer of the action-value function of $\pi_e$ [Wang et al., 2021], and 3) the original state-action features. Note that there is no representation *learning* involved for 2) and 3). We set the learning rate for all neural network training (encoder and FQE) to be the same, hyperparameter sweep ROPE across $\beta$ and the dimension of ROPE's encoder output, and report the lowest RMAE achieved at the end of FQE training. For hyperparameter sensitivity results, see Section 4.2.3. For training details, see Appendix D.

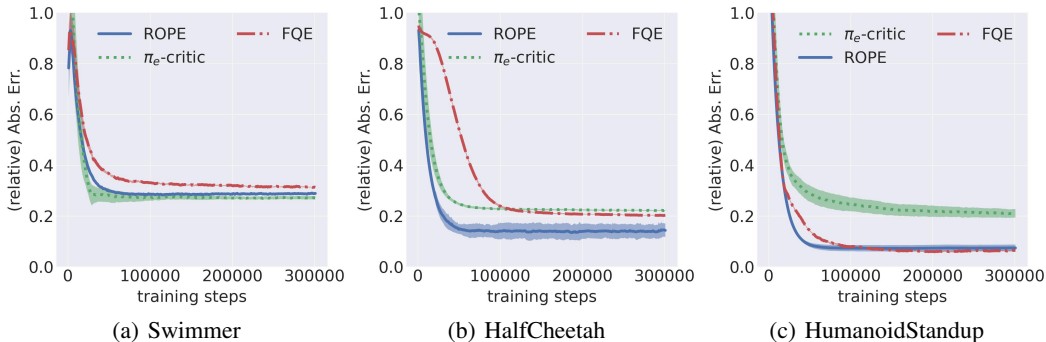

|  | (a) Swimmer | (b) HalfCheetah | (c) HumanoidStandup |

Figure 2: RMAE vs. training iterations of FQE on the custom datasets. IQM of errors for each domain were computed over 20 trials with 95% confidence intervals. Lower is better.

| | Algorithm | | |
| Dataset | BCRL | FQE | ROPE (ours) |
| --- | --- | --- | --- |
| HalfCheetah-random | $0.979 \pm 0.000$ | $\mathbf{0.807 \pm 0.010}$ | $0.990 \pm 0.001$ |
| HalfCheetah-medium | $0.830 \pm 0.007$ | $0.770 \pm 0.007$ | $\mathbf{0.247 \pm 0.001}$ |
| HalfCheetah-medium-expert | $0.685 \pm 0.013$ | $0.374 \pm 0.001$ | $\mathbf{0.078 \pm 0.043}$ |
| Walker2D-random | $1.022 \pm 0.001$ | Diverged | $\mathbf{0.879 \pm 0.009}$ |
| Walker2D-medium | $0.953 \pm 0.019$ | Diverged | $\mathbf{0.462 \pm 0.093}$ |
| Walker2D-medium-expert | $0.962 \pm 0.037$ | Diverged | $\mathbf{0.252 \pm 0.126}$ |
| Hopper-random | Diverged | Diverged | $\mathbf{0.680 \pm 0.05}$ |
| Hopper-medium | $61.223 \pm 92.282$ | Diverged | $\mathbf{0.208 \pm 0.048}$ |
| Hopper-medium-expert | $9.08 \pm 4.795$ | Diverged | $\mathbf{0.192 \pm 0.055}$ |

Table 1: Lowest RMAE achieved by algorithm on D4RL datasets. IQM of errors for each domain were computed over 20 trials with 95% confidence intervals. Algorithms that diverged had a significantly high final error and/or upward error trend (see Appendix D for training curves). Lower is better.

We find that FQE converges to an estimate of $\rho(\pi_e)$ when it is fed these different state-action features. We also see that when FQE is fed features from ROPE it produces more data-efficient OPE estimates than vanilla FQE. Under these mild distribution shift settings, $\pi_e$-critic also performs well since the output of the penultimate layer of $\pi_e$'s action-value function should have sufficient information to accurately estimate the action-value function of $\pi_e$.

**D4RL Dataset Results**   On the D4RL datasets, we analyze the final performance achieved by representation learning + OPE algorithms on datasets with varying distribution shift. In addition to the earlier baselines, we evaluate Bellman Complete Learning Representations (BCRL) [Chang et al., 2022], which learns linearly Bellman complete representations and produces an OPE estimate with Least-Squares Policy Evaluation (LSPE) instead of FQE. We could not evaluate $\pi_e$-critic since the D4RL $\pi_e$ critics were unavailable[3]. For BCRL, we use the publicly available code [4]. For a fair comparison, we hyperparameter tune the representation output dimension and encoder architecture size of BCRL. We hyperparameter tune ROPE the same way as done for the custom datasets. We set the learning rate for all neural network training (encoder and FQE) to be the same. In Table 1, we report the lowest RMAE achieved at the end of the OPE algorithm's training. For the corresponding training graphs, see Appendix D.

We find that ROPE improves the data-efficiency of FQE substantially across varying distribution shifts. BCRL performs competitively, but its poorer OPE estimates compared to ROPE is unsurprising since it is not designed for data-efficiency. It is also known that BCRL may produce less accurate OPE estimates compared to FQE [Chang et al., 2022]. FQE performs substantially worse on some datasets;

---

[3]https://github.com/google-research/deep_ope
[4]https://github.com/CausalML/bcrl

however, it is known that FQE can diverge under extreme distribution shift [Wang et al., 2020, 2021]. It is interesting, however, that ROPE is robust in these settings. We observe this robustness across a wide range of hyperparameters as well (see Section 4.2.3). We also find that when there is low diversity of rewards in the batch (for example, in the random datasets), it is more likely that the short-term distance component of ROPE is close to 0, which can result in a representation collapse.

### 4.2.3 Ablations

Towards a deeper understanding of ROPE, we now present an ablation study of ROPE.

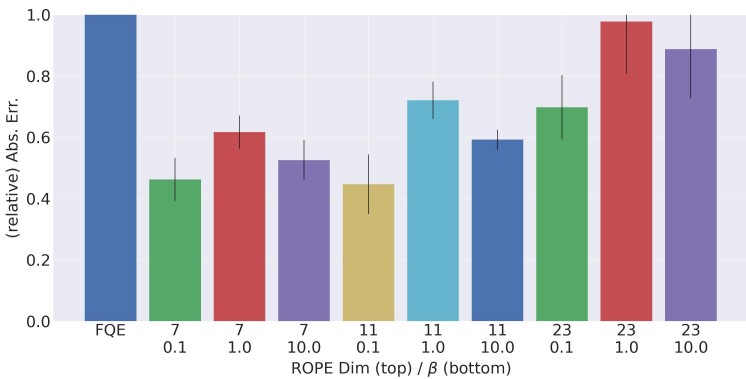

Figure 3: Hyperparameter sensitivity. FQE vs. ROPE when varying ROPE's encoder output dimension (top) and $\beta$ (bottom) on the Walker2D-medium D4RL dataset. IQM of errors are computed over 20 trials with 95% confidence intervals. Lower is better.

**Hyperparameter Sensitivity** In OPE, hyperparameter tuning with respect to RMAE is difficult since $\rho(\pi_e)$ is unknown in practice [Paine et al., 2020]. Therefore, we need OPE algorithms to not only produce accurate OPE estimates, but also to be robust to hyperparameter tuning. Specifically, we investigate whether ROPE's representations produce more data-efficient OPE estimates over FQE across ROPE's hyperparameters. In this experiment, we set the action-value function's learning rate to be the same for both algorithms. The hyperparameters for ROPE are: 1) the output dimension of the encoder and 2) $\beta$, the weight on the angular distance between encodings. We plot the results in Figure 3 and observe that ROPE is able to produce substantially more data-efficient estimates compared to FQE for a wide range of its hyperparameters on the Walker2D-medium dataset, where FQE diverged (see Table 1). While it is unclear what the optimal hyperparameters should be, we find similar levels of robustness on other datasets as well (see Appendix D).

**ROPE Representations Mitigate FQE Divergence** It has been shown theoretically [Wang et al., 2020] and empirically [Wang et al., 2021] that under extreme distribution shift, FQE diverges i.e. it produces OPE estimates that have arbitrarily large error. In Table 1, we also see similar results where FQE produces very high error on some datasets. FQE tends to diverge due to the deadly triad [Sutton and Barto, 2018]: 1) off-policy data, 2) bootstrapping, and 3) function approximation.

A rather surprising but encouraging result that we find is that even though ROPE faces the deadly triad, it produces representations that *significantly* mitigate FQE's divergence across a large number of trials and hyperparameter variations. To investigate how much ROPE aids convergence, we provide the performance profile[5] [Agarwal et al., 2021b] based on the RMAE distribution plot in Figure 4. Across all trials and hyperparameters, we plot the fraction of times an algorithm achieved an error less than some threshold. In addition to the earlier baselines, we also plot the performance of 1) FQE-CLIP which is FQE but whose bootstrapping targets are clipped between $\left[\frac{r_{\min}}{1-\gamma}, \frac{r_{\max}}{1-\gamma}\right]$, where $r_{\min}$ and $r_{\max}$ are the minimum and maximum rewards in the fixed dataset; and 2) FQE-DEEP, which is regular FQE but whose action-value function network is double the capacity of FQE (see Appendix D for specifics).

From Figure 4, we see that nearly $\approx 100\%$ of the runs of ROPE achieve an RMAE of $\leq 2$, while none of the FQE and FQE-DEEP runs produce even $\leq 10$ RMAE. The failure of FQE-DEEP suggests that the

---

[5]https://github.com/google-research/rliable/tree/master

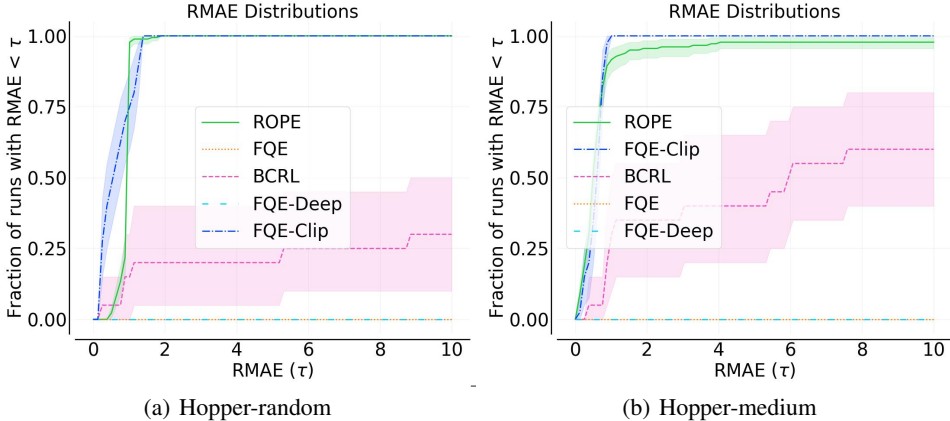

(a) Hopper-random    (b) Hopper-medium

Figure 4: RMAE distributions across all runs and hyperparameters for each algorithm, resulting in $\geq 20$ runs for each algorithm. The shaded region is a $95\%$ confidence interval. Larger area under the curve is better. For visualization, we cut off the horizontal axis at $10$ RMAE. FQE and FQE-DEEP are flat at $0$ i.e. neither had runs that produced an error less than $10$.

extra capacity ROPE has over FQE (since ROPE has its own neural network encoder) is insufficient to explain why ROPE produces accurate OPE estimates. We also find that in order to use FQE with the native state-action representations, it is necessary to use domain knowledge and clip the bootstrapped target. While FQE-CLIP avoids divergence, it is very unstable during training (see Appendix D). ROPE's ability to produce stable learning in FQE without any clipping is promising since it suggests that it is possible to improve the robustness of FQE if an appropriate representation is learned.

## 5    Limitations and Future Work

In this work, we showed that ROPE was able to improve the data-efficiency of FQE and produce lower-error OPE estimates than other OPE-based representations. Here, we highlight limitations and opportunities for future work. A limitation of ROPE and other bisimulation-based metrics is that if the diversity of rewards in the dataset is low, they are susceptible to representation collapse since the short-term distance is close to $0$. Further investigation is needed to determine how to overcome this limitation. Another very interesting future direction is to understand why ROPE's representations significantly mitigated FQE's divergence. A starting point would be to explore potential connections between ROPE and Bellman complete representations [Szepesvári and Munos, 2005] and other forms of representation regularizers for FQE[6].

## 6    Conclusion

In this paper we studied the challenge of pre-training representations to increase the data efficiency of the FQE OPE estimator. Inspired by work that learns state similarity metrics for control, we introduced ROPE, a new diffuse metric for measuring behavioral similarity between state-action pairs for OPE and used ROPE to learn state-action representations using available offline data. We theoretically showed that ROPE: 1) bounds the difference between the action-values between different state-action pairs and 2) results in bounded error between the value of $\pi_e$ according to the ground action-value and the action-value function that is fed with ROPE representations as input. We empirically showed that ROPE boosts the data-efficiency of FQE and achieves lower OPE error than other OPE-based representation learning algorithms. Finally, we conducted a thorough ablation study and showed that ROPE is robust to hyperparameter tuning and *significantly* mitigates FQE's divergence, which is a well-known challenge in OPE. To the best of our knowledge, our work is the first that successfully uses representation learning to improve the data-efficiency of OPE.

---

[6]`https://offline-rl-neurips.github.io/2021/pdf/17.pdf`

## Remarks on Negative Societal Impact

Our work is largely focused on studying fundamental RL research questions, and thus we do not see any immediate negative societal impacts. The aim of our work is to enable effective OPE in many real world domains. Effective OPE means that a user can estimate policy performance prior to deployment which can help avoid deployment of poor policies and thus positively impact society.

## Acknowledgments

Thanks to Adam Labiosa and the anonymous reviewers for feedback that greatly improved our work. Support for this research was provided by American Family Insurance through a research partnership with the University of Wisconsin—Madison's Data Science Institute.

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

# A  Theoretical Background

In this section, we include relevant background material.

**Definition 2** (Metric). *A metric, $d : X \times X \to \mathbb{R}_{\geq 0}$ has the following properties for some $x, y, z \in X$:*

1. *$d(x, x) = 0$*

2. *$d(x, y) = 0 \iff x = y$*

3. *Symmetry: $d(x, y) = d(y, x)$*

4. *Triangle inequality: $d(x, z) \leq d(x, y) + d(y, z)$*

**Definition 3** (Pseudo Metric). *A pseudo metric, $d : X \times X \to \mathbb{R}_{\geq 0}$ has the following properties for some $x, y, z \in X$:*

1. *$d(x, x) = 0$*

2. *Symmetry: $d(x, y) = d(y, x)$*

3. *Triangle inequality: $d(x, z) \leq d(x, y) + d(y, z)$*

*Crucially, a pseudo metric differs from a metric in that if $d(x, y) = 0$ it may be the case that $x \neq y$.*

**Definition 4** (Diffuse Metric). *A diffuse metric, $d : X \times X \to \mathbb{R}_{\geq 0}$ has the following properties for some $x, y, z \in X$:*

1. *$d(x, x) \geq 0$*

2. *Symmetry: $d(x, y) = d(y, x)$*

3. *Triangle inequality: $d(x, z) \leq d(x, y) + d(y, z)$*

*Crucially, a diffuse metric differs from a pseudo metric in that self-distances may be non-zero.*

For readers interested in distances that admit non-zero self-distances, we refer them to material on *partial metrics* [Matthews, 1992]. We make the following note as Castro et al. [2022]: the original definition of partial metrics (see Matthews [1992]) uses a different triangle inequality criterion than the one in Definition 4 and is too strict (i.e. diffuse metrics violate this triangle inequality criterion), so we consider the diffuse metric definition presented in this paper.

We now present background material on the Wasserstein and related distances.

**Definition 5** (Wasserstein Distance [Villani, 2008]). *Let $d : X \times X \to \mathbb{R}_{\geq 0}$ be a distance function and $\Omega$ the set of all joint distributions with marginals $\mu$ and $\lambda$ over the space $X$, then we have:*

$$W(d)(\mu, \lambda) = \left( \inf_{\omega \in \Omega} \mathbb{E}_{x_1, x_2 \sim \omega}[d(x_1, x_2)] \right) \tag{4}$$

**Definition 6** (Dual formulation of the Wasserstein Distance [Villani, 2008]). *Let $d : X \times X \to \mathbb{R}_{\geq 0}$ be a distance function and marginals $\mu$ and $\lambda$ over the space $X$, then we have:*

$$W(d)(\mu, \lambda) = \sup_{f \in Lip_{1,d}(X)} \mathbb{E}_{x_1 \sim \mu}[f(x_1)] - \mathbb{E}_{x_2 \sim \lambda}[f(x_2)] \tag{5}$$

*where $Lip_{1,d}(X)$ denotes the $1-$Lipschitz functions $f : X \to \mathbb{R}$ such that $|f(x_1) - f(x_2)| \leq d(x_1, x_2)$.*

**Definition 7** (Łukaszyk–Karmowski distance [Łukaszyk, 2004]). *Let $d : X \times X \to \mathbb{R}_{\geq 0}$ be a distance function and marginals $\mu$ and $\lambda$ over the space $X$, then we have:*

$$D_{LK}(d)(\mu, \lambda) = (\mathbb{E}_{x_1 \sim \mu, x_2 \sim \lambda}[d(x_1, x_2)]) \tag{6}$$

We then have the following fact: $W(d)(\mu, \lambda) \leq D_{LK}(d)(\mu, \lambda)$ i.e. the Wasserstein distance is upper-bounded by the Łukaszyk–Karmowski distance [Castro et al., 2022].

## B  Theoretical Results

**Proposition 1.** *The operator $\mathcal{F}^{\pi_e}$ is a contraction mapping on $\mathbb{R}^{\mathcal{X}\times\mathcal{X}}$ with respect to the $L^\infty$ norm.*

*Proof.* Consider $d, d' \in \mathbb{R}^{\mathcal{X}\times\mathcal{X}}$, then we have:

$$||(\mathcal{F}^{\pi_e}d)(s_1, a_1; s_2, a_2) - (\mathcal{F}^{\pi_e}d')(s_1, a_1; s_2, a_2)||_\infty$$
$$= ||\gamma \, \mathbb{E}_{s_1', s_2' \sim P, a_1', a_2' \sim \pi_e}[d(s_1', a_1'; s_2', a_2') - d'(s_1', a_1'; s_2', a_2')]||_\infty$$
$$= |\gamma| \cdot ||\, \mathbb{E}_{s_1', s_2' \sim P, a_1', a_2' \sim \pi_e}[d(s_1', a_1'; s_2', a_2') - d'(s_1', a_1'; s_2', a_2')]||_\infty$$
$$\leq \gamma \max_{s_1', a_1', s_2', a_2'} |d(s_1', a_1'; s_2', a_2') - d'(s_1', a_1'; s_2', a_2')]| = \gamma||d - d'||_\infty$$

$\square$

**Proposition 2.** *The operator $\mathcal{F}^{\pi_e}$ has a unique fixed point $d_{\pi_e} \in \mathbb{R}^{\mathcal{X}\times\mathcal{X}}$. Let $d_0 \in \mathbb{R}^{\mathcal{X}\times\mathcal{X}}$, then $\lim_{t\to\infty} \mathcal{F}^{\pi_e}_t(d_0) = d_{\pi_e}$.*

*Proof.* Since $\mathcal{F}^{\pi_e}$ is a contraction mapping and that $\mathbb{R}^{\mathcal{X}\times\mathcal{X}}$ is complete under the $L^\infty$ norm, by Banach's fixed-point theorem, $\lim_{t\to\infty} \mathcal{F}^{\pi_e}_t(d) = d_{\pi_e}$. $\square$

**Proposition 3.** *$d_{\pi_e}$ is a diffuse metric.*

*Proof.* To prove that $d_{\pi_e}$ is a diffuse metric, we need to show it has the following properties for $(s_1, a_1), (s_2, a_2), (s_3, a_3) \in \mathcal{X}$. We follow Castro et al. [2022]'s strategy (see Proposition 4.10) to prove that a distance function is a diffuse metric. Recall that $d_{\pi_e}(s_1, a_1; s_2, a_2) := |r(s_1, a_1) - r(s_2, a_2)| + \gamma \, \mathbb{E}_{s_1', s_2' \sim P, a_1', a_2' \sim \pi_e}[d_{\pi_e}(s_1', a_1'; s_2', a_2')]$.

1. Non-negativity i.e. $d_{\pi_e}(s_1, a_1; s_2, a_2) \geq 0$. Since $|r(s_1, a_1) - r(s_2, a_2)| \geq 0$, recursively rolling out the definition of $d_{\pi_e}$ means that $d_{\pi_e}(s_1, a_1; s_2, a_2)$ is a sum of discounted non-negative terms.

2. Symmetry i.e. $d_{\pi_e}(s_1, a_1; s_2, a_2) = d_{\pi_e}(s_2, a_2; s_1, a_1)$. Since $|r(s_1, a_1) - r(s_2, a_2)| = |r(s_2, a_2) - r(s_1, a_1)|$, unrolling $d_{\pi_e}(s_1, a_1; s_2, a_2)$ and $d_{\pi_e}(s_2, a_2; s_1, a_1)$ recursively results in the discounted sum of the same terms.

3. Triangle inequality i.e. $d_{\pi_e}(s_1, a_1; s_2, a_2) \leq d_{\pi_e}(s_1, a_1; s_3, a_3) + d_{\pi_e}(s_2, a_2; s_3, a_3)$. To show this fact, we will first consider an initialization to the distance function $d_0(s_1, a_1; s_2, a_2) = 0, \forall(s_1, a_1), (s_2, a_2) \in \mathcal{X}$ and consider repeated applications of the operator $\mathcal{F}^{\pi_e}$ to $d_0$, which we know will make $d_0$ converge to $d_{\pi_e}$ (Proposition 2). We will show by induction that each successive update $d_{t+1} = \mathcal{F}^{\pi_e}(d_t)$ satisfies the triangle inequality, which implies that $d_{\pi_e}$ satisfies the triangle inequality.

   We have the base the case at $t = 0$ trivially holding true due to the initialization of $d_0$. Now let the inductive hypothesis be true for all $t > 1$ i.e. $d_t(s_1, a_1; s_2, a_2) \leq d_t(s_1, a_1; s_3, a_3) + d_t(s_3, a_3; s_2, a_2)$ for any $(s_1, a_1), (s_2, a_2), (s_3, a_3) \in \mathcal{X}$. However, we know that:

$$d_{t+1}(s_1, a_1; s_2, a_2) = |r(s_1, a_1) - r(s_2, a_2)| + \gamma \, \mathbb{E}_{s_1', s_2' \sim P, a_1', a_2' \sim \pi_e}[d_t(s_1', a_1'; s_2', a_2')]$$

$$\overset{(a)}{=} |r(s_1, a_1) - r(s_2, a_2)| + r(s_3, a_3) - r(s_3, a_3) + \gamma \, \mathbb{E}_{s_1', s_2' \sim P, a_1', a_2' \sim \pi_e}[d_t(s_1', a_1'; s_2', a_2')]$$

$$\overset{(b)}{\leq} |r(s_1, a_1) - r(s_3, a_3)| + |r(s_2, a_2) - r(s_3, a_3)|$$
$$+ \gamma \, \mathbb{E}_{s_1', s_2' \sim P, a_1', a_2' \sim \pi_e}[d_t(s_1', a_1'; s_2', a_2')]$$

$$\overset{(c)}{\leq} |r(s_1, a_1) - r(s_3, a_3)| + |r(s_2, a_2) - r(s_3, a_3)|$$
$$+ \gamma \, \mathbb{E}_{s_1', s_2', s_3' \sim P, a_1', a_2', a_3' \sim \pi_e}[d_t(s_1', a_1'; s_3', a_3') + d_t(s_3', a_3'; s_2', a_2')]$$

$$= |r(s_1, a_1) - r(s_3, a_3)| + \gamma \, \mathbb{E}_{s_1', s_3' \sim P, a_1', a_3' \sim \pi_e}[d_t(s_1', a_1'; s_3', a_3')]$$
$$+ |r(s_2, a_2) - r(s_3, a_3)| + \gamma \, \mathbb{E}_{s_2', s_3' \sim P, a_2', a_3' \sim \pi_e}[d_t(s_3', a_3'; s_2', a_2')]$$

$$= d_{t+1}(s_1, a_1; s_3, a_3) + d_{t+1}(s_2, a_2; s_3, a_3)$$

$$d_{t+1}(s_1, a_1; s_2, a_2) \leq d_{t+1}(s_1, a_1; s_3, a_3) + d_{t+1}(s_2, a_2; s_3, a_3)$$

where (a) is due to adding and subtracting $r(s_3, a_3)$, (b) is due to Jensen's inequality, (c) is due to application of the inductive hypothesis. Thus, the triangle inequality is satisfied for all $t \geq 0$, and given that $d_{t+1} \to d_{\pi_e}$, we have that $d_{\pi_e}$ also satisfies the triangle inequality.

$\square$

**Theorem 1.** *For any evaluation policy $\pi_e$ and $(s_1, a_1), (s_2, a_2) \in \mathcal{X}$, we have that $|q^{\pi_e}(s_1, a_1) - q^{\pi_e}(s_2, a_2)| \leq d_{\pi_e}(s_1, a_1, ; s_2, a_2)$.*

*Proof.* To prove this fact, we follow Castro et al. [2022] (see Proposition 4.8) and use a co-inductive argument [Kozen, 2006]. We will show that if $|q^{\pi_e}(s_1, a_1) - q^{\pi_e}(s_2, a_2)| \leq d(s_1, a_1, ; s_2, a_2)$ holds true for some specific symmetric $d \in \mathbb{R}^{\mathcal{X} \times \mathcal{X}}$, then the statement also holds true for $\mathcal{F}^{\pi_e}(d)$, which means it will hold for $d_{\pi_e}$.

We have that for any $(s, a) \in \mathcal{X}$, $\max_{s,a} \frac{-|r(s,a)|}{1-\gamma} \leq q^{\pi_e}(s, a) \leq \max_{s,a} \frac{|r(s,a)|}{1-\gamma}$. Thus, for any $(s_1, a_1), (s_2, a_2) \in \mathcal{X}$, we have that $|q^{\pi_e}(s_1, a_1) - q^{\pi_e}(s_2, a_2)| \leq 2 \max_{s,a} \frac{|r(s,a)|}{1-\gamma}$. We can then assume that our specific symmetric $d$ is the constant function $d(s_1, a_1; s_2, a_2) = 2 \max_{s,a} \frac{|r(s,a)|}{1-\gamma}$, which satisfies our requirement that $|q^{\pi_e}(s_1, a_1) - q^{\pi_e}(s_2, a_2)| \leq d(s_1, a_1, ; s_2, a_2)$.

Therefore, we have $q^{\pi_e}(s_1, a_1) - q^{\pi_e}(s_2, a_2)$

$$= r(s_1, a_1) - r(s_2, a_2) + \gamma \sum_{s_1' \in \mathcal{S}} \sum_{a_1' \in \mathcal{A}} P(s_1'|s_1, a_1)\pi_e(a_1'|s_1')q^{\pi_e}(s_1', a_1') - \gamma \sum_{s_2' \in \mathcal{S}} \sum_{a_2' \in \mathcal{A}} P(s_2'|s_2, a_2)\pi_e(a_2'|s_2')q^{\pi_e}(s_2', a_2')$$

$$\leq |r(s_1, a_1) - r(s_2, a_2)| + \gamma \sum_{s_1', s_2' \in \mathcal{S}} \sum_{a_1', a_2' \in \mathcal{A}} P(s_1'|s_1, a_1)\pi_e(a_1'|s_1')P(s_2'|s_2, a_2)\pi_e(a_2'|s_2')(q^{\pi_e}(s_1', a_1') - q^{\pi_e}(s_2', a_2'))$$

$$\overset{(a)}{\leq} |r(s_1, a_1) - r(s_2, a_2)| + \gamma \sum_{s_1', s_2' \in \mathcal{S}} \sum_{a_1', a_2' \in \mathcal{A}} P(s_1'|s_1, a_1)\pi_e(a_1'|s_1')P(s_2'|s_2, a_2)\pi_e(a_2'|s_2')d(s_1', a_1'; s_2', a_2')$$

$$= \mathcal{F}^{\pi_e}(d)(s_1, a_1; s_2, a_2)$$

where (a) follows from the induction hypothesis. Similarly, by symmetry, we can show that $q^{\pi_e}(s_2, a_2) - q^{\pi_e}(s_1, a_1) \leq \mathcal{F}^{\pi_e}(d)(s_1, a_1; s_2, a_2)$. Thus, we have it that $|q^{\pi_e}(s_1, a_1) - q^{\pi_e}(s_2, a_2)| \leq d_{\pi_e}(s_1, a_1, ; s_2, a_2)$. $\square$

**Lemma 1.** *Assume the rewards $\mathcal{R} : \mathcal{S} \times \mathcal{A} \to \Delta([0, 1])$ then given an aggregated MDP $\widetilde{\mathcal{M}} = \langle \widetilde{\mathcal{S}}, \widetilde{\mathcal{A}}, \widetilde{\mathcal{R}}, \widetilde{P}, \gamma, \tilde{d}_0 \rangle$ constructed by aggregating state-actions in an $\epsilon$-neighborhood based on $d_{\pi_e}$, and an encoder $\phi : \mathcal{X} \to \widetilde{\mathcal{X}}$ that maps state-actions in $\mathcal{X}$ to these clusters, the action-value for the evaluation policy $\pi_e$ in the two MDPs are bounded as:*

$$|q^{\pi_e}(x) - \tilde{q}^{\pi_e}(\phi(x))| \leq \frac{2\epsilon}{(1-\gamma)}$$

*Proof.* The proof closely follows that of Lemma 8 of Kemertas and Aumentado-Armstrong [2021], which is in turn based on Theorem 5.1 of Ferns et al. [2004]. The main difference between their theorems and ours is that the former is based on state representations and the latter is based on optimal state-value functions, while ours is focused on state-action representations for $\pi_e$.

We first remark that this new aggregated MDP, $\widetilde{\mathcal{M}}$, can be viewed as a Markov reward process (MRP) where the "states" are aggregated state-action pairs of the original MDP, $\mathcal{M}$. We now define the reward function and transition dynamics of the clustered MRP $\widetilde{\mathcal{M}}$, where $|\phi(x)|$ is the size of the cluster $\phi(x)$. Note that $\mathbb{P}$ denotes the probability of the event.

$$\tilde{r}(\phi(x)) = \frac{1}{|\phi(x)|} \sum_{y \in \phi(x)} r(y)$$

$$\widetilde{P}(\phi(x')|\phi(x)) = \frac{1}{|\phi(x)|} \sum_{y \in \phi(x)} \mathbb{P}(\phi(x')|y)$$

Then we have: $|q^{\pi_e}(x) - \tilde{q}^{\pi_e}(\phi(x))|$

$$= \left| r(x) - \tilde{r}(\phi(x)) + \gamma \sum_{x' \in \mathcal{X}} P(x'|x)q^{\pi_e}(x') - \gamma \sum_{\phi(x') \in \widetilde{\mathcal{X}}} \widetilde{P}(\phi(x')|\phi(x))\tilde{q}^{\pi_e}(\phi(x')) \right|$$

$$\stackrel{(a)}{=} \left| r(x) - \frac{1}{|\phi(x)|} \sum_{y \in \phi(x)} r(y) + \gamma \sum_{x' \in \mathcal{X}} P(x'|x)q^{\pi_e}(x') - \gamma \frac{1}{|\phi(x)|} \sum_{\phi(x') \in \widetilde{\mathcal{X}}} \sum_{y \in \phi(x)} \mathbb{P}(\phi(x')|y)\tilde{q}^{\pi_e}(\phi(x')) \right|$$

$$\stackrel{(b)}{=} \frac{1}{|\phi(x)|} \left| |\phi(x)|r(x) - \sum_{y \in \phi(x)} r(y) + \gamma|\phi(x)| \sum_{x' \in \mathcal{X}} P(x'|x)q^{\pi_e}(x') - \gamma \sum_{\phi(x') \in \widetilde{\mathcal{X}}} \sum_{y \in \phi(x)} \mathbb{P}(\phi(x')|y)\tilde{q}^{\pi_e}(\phi(x')) \right|$$

$$\stackrel{(c)}{=} \frac{1}{|\phi(x)|} \left| \sum_{y \in \phi(x)} (r(x) - r(y)) + \sum_{y \in \phi(x)} \left( \gamma \sum_{x' \in \mathcal{X}} P(x'|x)q^{\pi_e}(x') - \gamma \sum_{\phi(x') \in \widetilde{\mathcal{X}}} \mathbb{P}(\phi(x')|y)\tilde{q}^{\pi_e}(\phi(x')) \right) \right|$$

$$\stackrel{(d.1)}{\leq} \frac{1}{|\phi(x)|} \sum_{y \in \phi(x)} \left( |r(x) - r(y)| + \gamma \left| \sum_{x' \in \mathcal{X}} P(x'|x)q^{\pi_e}(x') - \sum_{\phi(x') \in \widetilde{\mathcal{X}}} \mathbb{P}(\phi(x')|y)\tilde{q}^{\pi_e}(\phi(x')) \right| \right)$$

$$\stackrel{(d.2)}{=} \frac{1}{|\phi(x)|} \sum_{y \in \phi(x)} \left( |r(x) - r(y)| + \gamma \left| \sum_{x' \in \mathcal{X}} P(x'|x)q^{\pi_e}(x') - \sum_{\phi(x') \in \widetilde{\mathcal{X}}} \sum_{z \in \phi(x')} P(z|y)\tilde{q}^{\pi_e}(\phi(x')) \right| \right)$$

$$\stackrel{(d.3)}{=} \frac{1}{|\phi(x)|} \sum_{y \in \phi(x)} \left( |r(x) - r(y)| + \gamma \left| \sum_{x' \in \mathcal{X}} P(x'|x)q^{\pi_e}(x') - \sum_{x' \in \mathcal{X}} P(x'|y)\tilde{q}^{\pi_e}(\phi(x')) \right| \right)$$

$$\stackrel{(e)}{\leq} \frac{1}{|\phi(x)|} \sum_{y \in \phi(x)} \left( |r(x) - r(y)| + \gamma \left| \sum_{x' \in \mathcal{X}} (P(x'|x)q^{\pi_e}(x') - P(x'|y)\tilde{q}^{\pi_e}(\phi(x'))) \right| \right)$$

$$\stackrel{(f)}{\leq} \frac{1}{|\phi(x)|} \sum_{y \in \phi(x)} \left( |r(x) - r(y)| + \gamma \left| \sum_{x' \in \mathcal{X}} (P(x'|x)q^{\pi_e}(x') - P(x'|y)q^{\pi_e}(x')) \right| \right)$$

$$+ \frac{\gamma}{|\phi(x)|} \sum_{y \in \phi(x)} \left( \left| \sum_{x' \in \mathcal{X}} P(x'|y)(q^{\pi_e}(x') - \tilde{q}^{\pi_e}(\phi(x'))) \right| \right)$$

$$\stackrel{(g)}{\leq} \frac{1}{|\phi(x)|} \sum_{y \in \phi(x)} \left( |r(x) - r(y)| + \gamma \left| \sum_{x' \in \mathcal{X}} (P(x'|x) - P(x'|y)) q^{\pi_e}(x') \right| + \gamma \|q - \tilde{q}\|_\infty \right)$$

$$\stackrel{(h)}{=} \frac{1}{|\phi(x)|} \sum_{y \in \phi(x)} \left( |r(x) - r(y)| + \gamma \left| \mathbb{E}_{x' \sim P(\cdot|x)}[q^{\pi_e}(x')] - \mathbb{E}_{x' \sim P(\cdot|y)}[q^{\pi_e}(x')] \right| + \gamma \|q - \tilde{q}\|_\infty \right)$$

where (a) is due to the definition of $\tilde{r}$ and $\widetilde{P}$, (b) is due to multiplying and dividing by $|\phi(x)|$, (c) is due to re-arranging terms, (d.1) is due to Jensen's inequality, (d.2 and d.3) are disaggregating the sums over clustered state-actions into sums over original state-actions by expanding $\mathbb{P}(\phi(x')|y) = \sum_{x \in \phi(x')} P(x|y)$ for each clustered state-action, $\phi(x')$, (e) is grouping the terms, (f) is by adding and subtracting $\frac{1}{|\phi(x)|} \sum_{y \in \phi(x)} P(x'|y)q^{\pi_e}(x')$, (g) is since the infinity norm of the difference of the action-values is greater than the expected difference, (h) is re-writing the expression in terms of expectations.

From Theorem 1 we know $q^{\pi_e}$ is 1-Lipschitz with respect to the distance function $d_{\pi_e}$. Notice that (h) contains the dual formulation of the Wasserstein distance where $f = q^{\pi_e}$ (see Definition 6). We

can then re-write (h) in terms of original definition of the Wasserstein distance:

$$|q^{\pi_e}(x) - \tilde{q}^{\pi_e}(\phi(x))| \leq \frac{1}{|\phi(x)|} \sum_{y \in \phi(x)} \left( |r(x) - r(y)| + \gamma W(d_{\pi_e})(P(\cdot|x), P(\cdot|y)) + \gamma \|q - \tilde{q}\|_\infty \right)$$

$$\overset{(i)}{\leq} \frac{1}{|\phi(x)|} \sum_{y \in \phi(x)} \left( |r(x) - r(y)| + \gamma D_{\mathrm{LK}}(d_{\pi_e})(x', y') + \gamma \|q - \tilde{q}\|_\infty \right)$$

$$\overset{(j)}{=} \frac{1}{|\phi(x)|} \sum_{y \in \phi(x)} \left( |r(x) - r(y)| + \gamma \, \mathbb{E}_{x' \sim \mathbb{P}^{\pi_e}, y' \sim \mathbb{P}^{\pi_e}}[d_{\pi_e}(x', y')] + \gamma \|q - \tilde{q}\|_\infty \right)$$

$$\overset{(k)}{=} \frac{1}{|\phi(x)|} \sum_{y \in \phi(x)} \left( d_{\pi_e}(x, y) + \gamma \|q - \tilde{q}\|_\infty \right)$$

$$\overset{(l)}{\leq} 2\epsilon + \gamma \|q - \tilde{q}\|_\infty$$

$$|q^{\pi_e}(x) - \tilde{q}^{\pi_e}(\phi(x))| \overset{(m)}{\leq} \frac{2\epsilon}{1 - \gamma}, \forall x \in \mathcal{X}$$

where (i) is due the fact that the Łukaszyk–Karmowski, $D_{\mathrm{LK}}$, upper bounds the Wasserstein distance, (j) is using Definition 7, (k) is due to the definition of $d_{\pi_e}$, and (l) is due the fact that the maximum distance between any two $x, y \in \phi(x)$ is at most $2\epsilon$, which is greater than the average distance between any one point to every other point in the cluster, and (m) is due to $\|q - \tilde{q}\|_\infty \leq \frac{2\epsilon}{1-\gamma}$. $\qquad \square$

**Theorem 2.** *Under the same conditions as Lemma 1, the difference between the expected fitted q-evaluation (*FQE*) estimate and the expected estimate of* FQE+ROPE *is bounded:*

$$\left| \mathbb{E}_{s_0, a_0 \sim \pi_e}[q^{\pi_e}(s_0, a_0)] - \mathbb{E}_{s_0, a_0 \sim \pi_e}[q^{\pi_e}(\phi(s_0, a_0))] \right| \leq \frac{2\epsilon}{(1 - \gamma)}$$

*Proof.* From Lemma 1 we have that $|q^{\pi_e}(s_0, a_0) - q^{\pi_e}(\phi(s_0, a_0))| \leq \frac{2\epsilon}{(1-\gamma)}$.

$$\left| \mathbb{E}_{s_0, a_0 \sim \pi_e}[q^{\pi_e}(s_0, a_0)] - \mathbb{E}_{s_0, a_0 \sim \pi_e}[q^{\pi_e}(\phi(s_0, a_0))] \right| = \left| \mathbb{E}_{s_0, a_0 \sim \pi_e}[q^{\pi_e}(s_0, a_0) - q^{\pi_e}(\phi(s_0, a_0))] \right|$$

$$\overset{(a)}{\leq} \mathbb{E}_{s_0, a_0 \sim \pi_e}[|q^{\pi_e}(s_0, a_0) - q^{\pi_e}(\phi(s_0, a_0))|]$$

$$\overset{(b)}{\leq} \mathbb{E}_{s_0, a_0 \sim \pi_e}\left( \frac{2\epsilon}{1 - \gamma} \right)$$

$$= \frac{2\epsilon}{(1 - \gamma)},$$

where (a) follows from Jensen's inequality and (b) follows from Lemma 1. $\qquad \square$

## C  ROPE Pseudo-code

---
**Algorithm 1** ROPE+FQE
---
1: Input: policy to evaluate $\pi_e$, batch $\mathcal{D}$, encoder parameters class $\Omega$, action-value parameter class $\Xi$, encoder function $\phi : \mathcal{S} \times \mathcal{A} \to \mathbb{R}^d$, action-value function $q : \mathcal{S} \times \mathcal{A} \to \mathbb{R}$.
2: $\hat{\omega} := \arg\min_{\omega \in \Omega}$
$$\mathbb{E}_{(s_1, a_1, s_1'), (s_2, a_2, s_2') \sim \mathcal{D}} \left[ \rho \left( |r(s_1, a_1) - r(s_2, a_2)| + \gamma \, \mathbb{E}_{a_1', a_2' \sim \pi_e}[\tilde{d}_{\bar{\omega}}(s_1', a_1'; s_2', a_2')] - \tilde{d}_\omega(s_1, a_1; s_2, a_2) \right) \right]$$
{ROPE training phase; where $\tilde{d}_\omega(s_1, a_1; s_2, a_2) := \frac{\|\phi_\omega(s_1, a_1)\|_2^2 + \|\phi_\omega(s_2, a_2)\|_2^2}{2} + \beta\theta(\phi_\omega(s_1, a_1), \phi_\omega(s_2, a_2))$, $\bar{\omega}$ are fixed parameters of target network, and $\rho$ is the Huber loss. See Section 3.1 for more details.}
3: $\hat{\xi} := \arg\min_{\xi \in \Xi} \mathbb{E}_{(s, a, s') \sim \mathcal{D}} \left[ \rho \left( r(s, a) + \gamma \, \mathbb{E}_{a' \sim \pi_e}[q_{\bar{\xi}}(\phi_{\hat{\omega}}(s', a'))] - q_\xi(\phi_{\hat{\omega}}(s, a)) \right) \right]$ {FQE using fixed encoder $\phi_{\hat{\omega}}$ from Step 2, where $\rho$ is the Huber loss.}
4: Return $q_{\hat{\xi}}$ {Estimated action-value function of $\pi_e$, $q^{\pi_e}$.}

---

# D  Empirical Results

We now include additional experiments that were deferred from the main text.

## D.1  Gridworld Visualizations

In Section 4.2.1, we visualize how ROPE and on-policy MICO group state-actions pairs. We now consider two additional metrics that group state-action pairs:

1. Policy similarity metric [Agarwal et al., 2021a]: $d_{\text{PSM}}(s_1, a_1; s_2, a_2) := |\pi_{\text{e}}(a_1|s_1) - \pi_{\text{e}}(a_2|s_2)| + \gamma \mathbb{E}_{a_1', a_2' \sim \pi_{\text{e}}}[d_{\text{PSM}}((s_1', a_1'), (s_2', a_2'))]$. This metric measures short- and long-term similarity based on how $\pi_{\text{e}}$ acts in different states, not in terms of the rewards and returns it receives.

2. Random policy similarity metric [Dadashi et al., 2021]: $d_{\text{RAND}}(s_1, a_1; s_2, a_2) := |r(s_1, a_1) - r(s_2, a_2)| + \gamma \mathbb{E}_{a' \sim \mathcal{U}(\mathcal{A})}[d_{\text{RAND}}((s_1', a'), (s_2', a'))]$. Similar to $d_{\pi_{\text{e}}}$, but considers behavior of a random policy that samples actions uniformly.

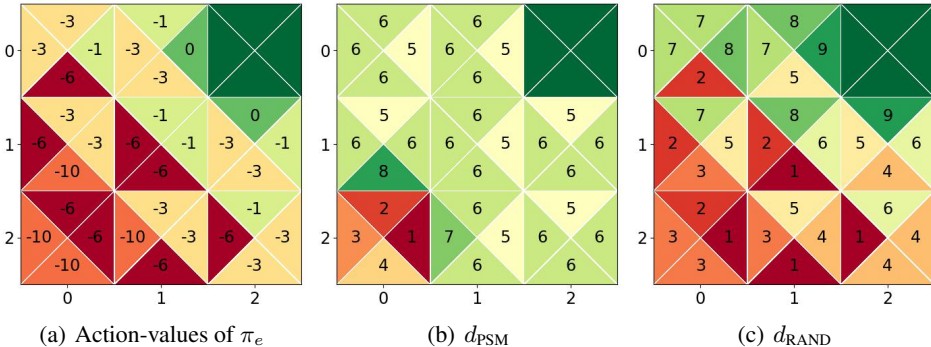

|  (a) Action-values of $\pi_e$  |  (b) $d_{\text{PSM}}$  |  (c) $d_{\text{RAND}}$  |

Figure 5: Figure (a): $q^{\pi_e}$ for $\pi_e$. Center and right: group clustering according to PSM (Figure (b)) and random-policy metric (Figure (c)) (center number in each triangle is group ID). Two state-action pairs are grouped together if their distance according to the specific metric is 0. The top right cell is blank since it is the terminal state, which is not grouped.

From Figure 5, we reach the same conclusion as we did in Section 4.2.1: that existing state-action similarity metrics are unsuitable for learning $q^{\pi_e}$ due to how they group state-action pairs.

## D.2  Deep OPE Experiments

We now present additional details on our empirical setup and additional experiments.

### D.2.1  Additional Empirical Setup Details

Before applying any of the algorithms, we normalize the states of the dataset to make the each feature dimension have 0 mean and 1 standard deviation.

**FQE Training Details**   In all experiments and all datasets, we use a neural network as FQE's action-value function with 2 layers and 256 neurons using RELU activation function. We use mini-batch gradient descent to train the FQE network with mini-batch sizes of 512 and for 300K gradient steps. We use the Adam optimizer with learning rate $1e^{-5}$ and weight decay $1e^{-2}$. FQE minimizes the Huber loss. The only changes for FQE-DEEP are that it uses a neural network size of 4 layers with 256 neurons and trains for 500K gradient steps. Preliminary results with lower learning rates such as $5e^{-6}$ and $1e^{-6}$ did not make a difference. FQE uses an exponentially-moving average target network with $\tau = 0.005$ updated every epoch.

**ROPE and BCRL Details**   In all experiments and datasets, we use a neural network as the state-action encoder for ROPE with 2 layers and 256 neurons with the RELU activation. We use mini-batch gradient descent to train the the encoder network with mini-batch sizes of 512 and for 300K

gradient steps. For ROPE and BCRL, we hyperparameter sweep the output dimension of the encoder. Additionally, for ROPE, we sweep over the angular distance scalar, $\beta$. For the output dimension, we sweep over dimensions: $\{|X|/3, |X|/2, |X|\}$, where $|X|$ is the dimension of the original state-action space of the environment. For $\beta$, we sweep over $\{0.1, 1, 10\}$. The best performing hyperparameter set is the one that results in lowest RMAE (from $\rho(\pi_e)$) at the end of FQE training. ROPE uses an exponentially-moving average target network with $\tau = 0.005$ updated every epoch. Finally, the output of ROPE's encoder is fed through a LayerNorm [Ba et al., 2016] layer, followed by a TANH layer. ROPE minimizes the Huber loss.

When computing $d^{\pi_e} \approx \tilde{d}_\omega$ ROPE uses the same procedure as MICO (appendix C.2. of Castro et al. [2022]):

$$\tilde{d}_\omega(s_1, a_1; s_2, a_2) \coloneqq \frac{||\phi_\omega(s_1, a_1)||_2^2 + ||\phi_{\bar{\omega}}(s_2, a_2)||_2^2}{2} + \beta\theta(\phi_\omega(s_1, a_1), \phi_{\bar{\omega}}(s_2, a_2))$$

where it applies the target network parameters, $\bar{\omega}$, on the $(s_2, a_2)$ pair for stability. For the angular distance $\theta(\phi_\omega(s_1, a_1), \phi_\omega(s_2, a_2))$, we have the cosine-similarity and the angle as below. Note in practice, for numerical stability, a small constant (e.g. $1e^{-6}$ or $5e^{-5}$) may have to be added when computing the square-root.

$$\text{CS}(\phi_\omega(s_1, a_1), \phi_\omega(s_2, a_2)) = \frac{\langle\phi_\omega(s_1, a_1), \phi_\omega(s_2, a_2)\rangle}{||\phi_\omega(s_1, a_1)||||\phi_\omega(s_2, a_2)||}$$

$$\theta(\phi_\omega(s_1, a_1), \phi_\omega(s_2, a_2)) = \text{arctan2}\left(\sqrt{1 - \text{CS}(\phi_\omega(s_1, a_1), \phi_\omega(s_2, a_2))^2}, \text{CS}(\phi_\omega(s_1, a_1), \phi_\omega(s_2, a_2))\right)$$

**Custom Datasets** We generate the datasets by training policies in the environment using SAC [Haarnoja et al., 2018] and take the final policy at the end of training as $\pi_e$ and we use an earlier policy with lower performance as the behavior policy. The expected discounted return of the policies and datasets for each domain is given in Table 2 ($\gamma = 0.99$). The values for the evaluation and behavior policies were computed by running each for 300 rollout trajectories, which was more than a sufficient amount for the estimate to converge, and averaging the discounted return (note that Chang et al. [2022] use 200 rollout trajectories).

|  | $\rho(\pi_e)$ | $\rho(\pi_b)$ |
|---|---|---|
| HumanoidStandup | 14500 | 13000 |
| Swimmer | 43 | 31 |
| HalfCheetah | 544 | 308 |

Table 2: Policy values of the evaluation policy and behavior policy.

**D4RL Datasets** Due to known discrepancy issues between newer environments of gym[7], we generat our datasets instead of using the publicly available ones. To generate the datasets, we use the publicly available policies [8]. For each domain, the expert and evaluation policy was the 10th (last policy) from training. The medium and behavior policy was the 5th policy. We added a noise of 0.1 to the policies.

### D.2.2 FQE Training Iteration Curves for D4RL Datasets

In this section, we include the remaining FQE training iteration curves (OPE error vs. gradient steps) for the D4RL dataset (Figure 6). We can see that FQE diverges in multiple settings while ROPE is very stable. While FQE-CLIP does not diverge, it is still highly unstable.

### D.2.3 Ablation: ROPE Hyperparameter Sensitivity

Similar to the results in Section 4.2.3, we show ROPE's hyperparameter sensitivity on all the custom and D4RL datasets. In general, we find that ROPE is robust to hyperparameter tuning, and it produces more data-efficient OPE estimates than FQE for a wide variety of its hyperparameters. See Figures 7 to 10.

---

[7]https://github.com/Farama-Foundation/D4RL/tree/master
[8]https://github.com/google-research/deep_ope

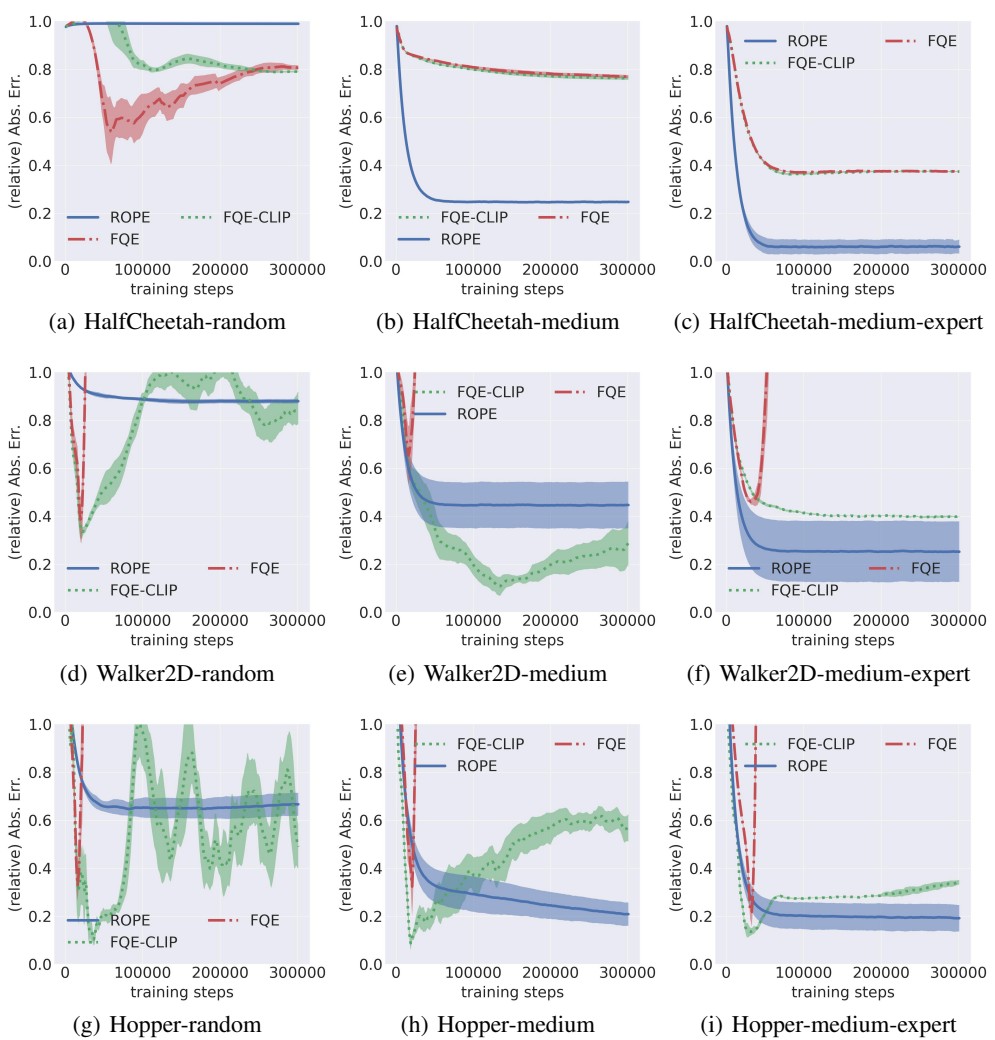

Figure 6: RMAE vs. training iterations of FQE on the D4RL datasets. IQM of errors for each domain were computed over 20 trials with 95% confidence intervals. Lower is better.

Note that in the bar graphs, we limit the vertical axis to 1. In the Hopper and Walker D4RL experiments, FQE diverged and had an error significantly larger than 1.

### D.2.4 Ablation: RMAE Distributions

In this section, show the remaining RMAE distribution curves [Agarwal et al., 2021b] of each algorithm on all datasets. We reach the similar conclusion that on very difficult datasets, ROPE significantly mitigates the divergence of FQE and that to avoid FQE divergence it is necessary to clip the bootstrapping target. See Figures 11 to 14.

### D.2.5 Training Loss Curves for ROPE and FQE

In this section, we include the training loss curves for ROPE's training, FQE's training using ROPE representations as input, and normal FQE and FQE-CLIP. The training curves are a function of the algorithms hyperparameters (learning rate for FQE, $\beta$ and representation output dimension for ROPE). We can see that on difficult datasets, the loss of FQE diverges. On the other hand, with ROPE, FQE's divergence is significantly mitigated. Note that ROPE does not eliminate the divergence. See Figures 15 to 18.

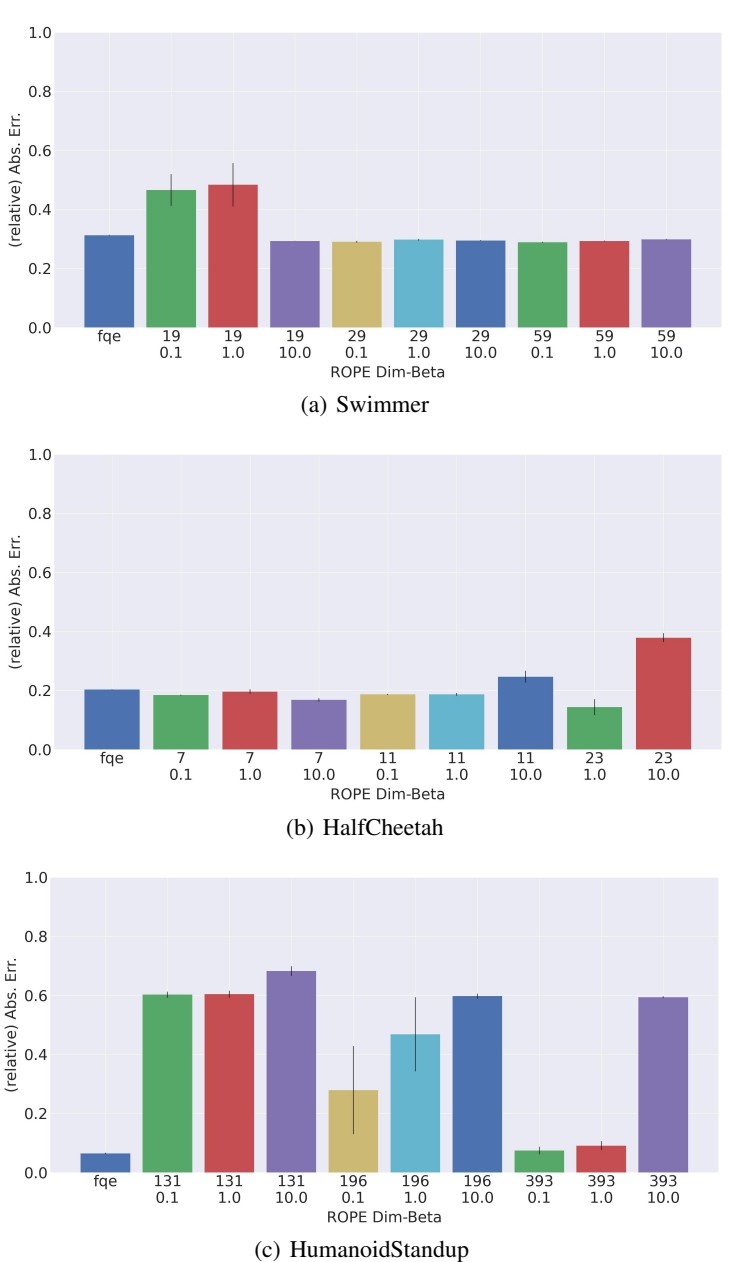

(a) Swimmer

(b) HalfCheetah

(c) HumanoidStandup

Figure 7: FQE vs. ROPE when varying ROPE's encoder output dimension (top) and $\beta$ (bottom) on the custom datasets. IQM of errors are computed over 20 trials with $95\%$ confidence intervals. Lower is better.

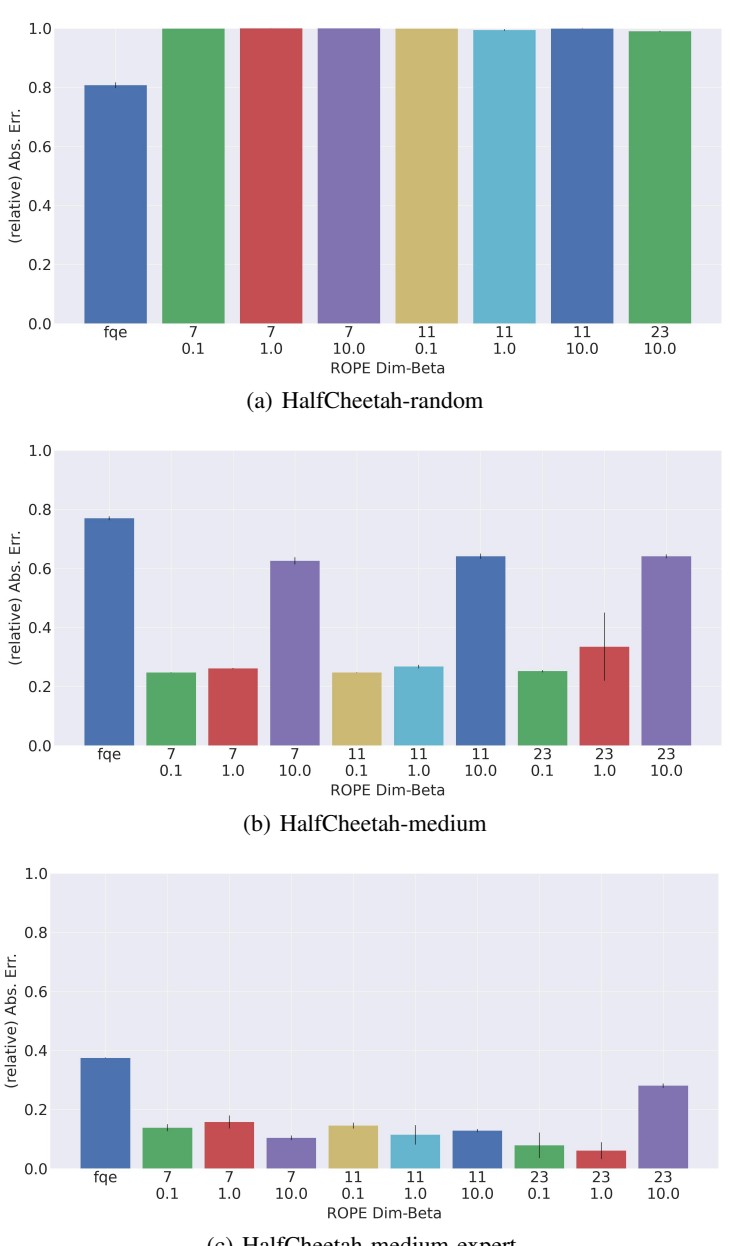

(a) HalfCheetah-random

(b) HalfCheetah-medium

(c) HalfCheetah-medium-expert

Figure 8: FQE vs. ROPE when varying ROPE's encoder output dimension (top) and $\beta$ (bottom) on the D4RL datasets. IQM of errors are computed over 20 trials with 95% confidence intervals. Lower is better.

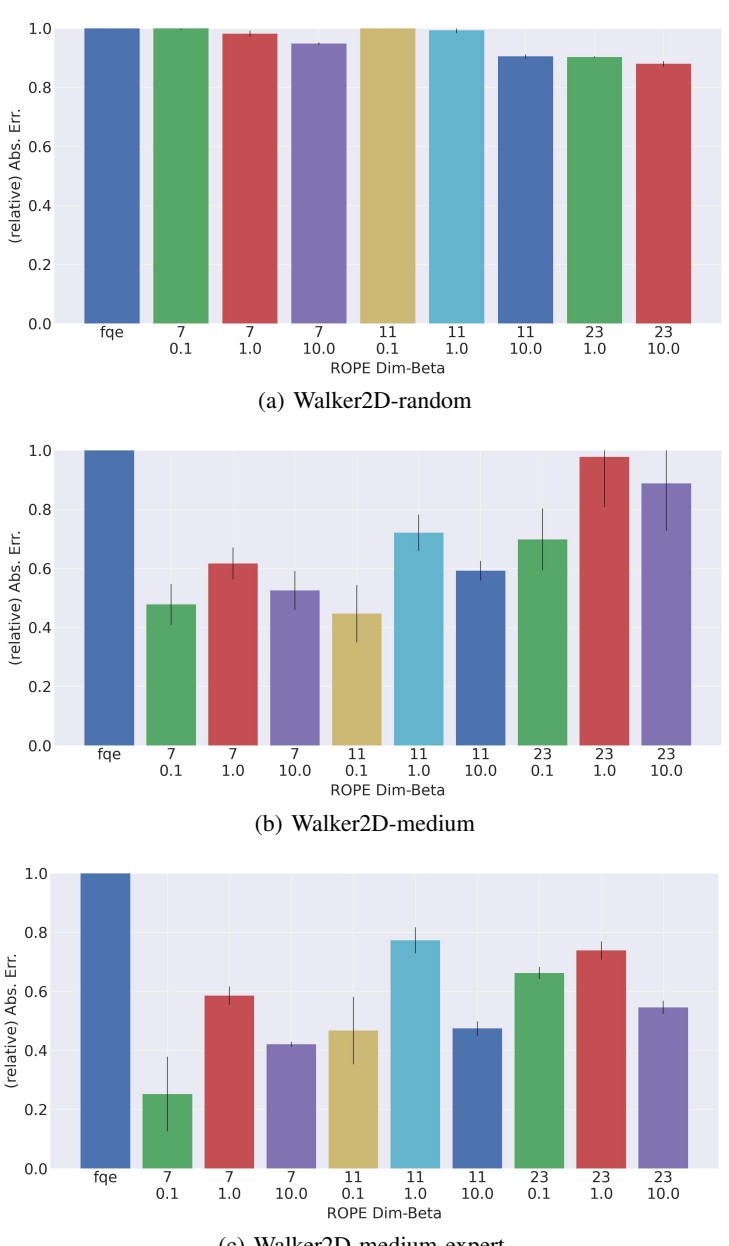

(a) Walker2D-random

(b) Walker2D-medium

(c) Walker2D-medium-expert

Figure 9: FQE vs. ROPE when varying ROPE's encoder output dimension (top) and $\beta$ (bottom) on the D4RL datasets. IQM of errors are computed over 20 trials with 95% confidence intervals. Lower is better.

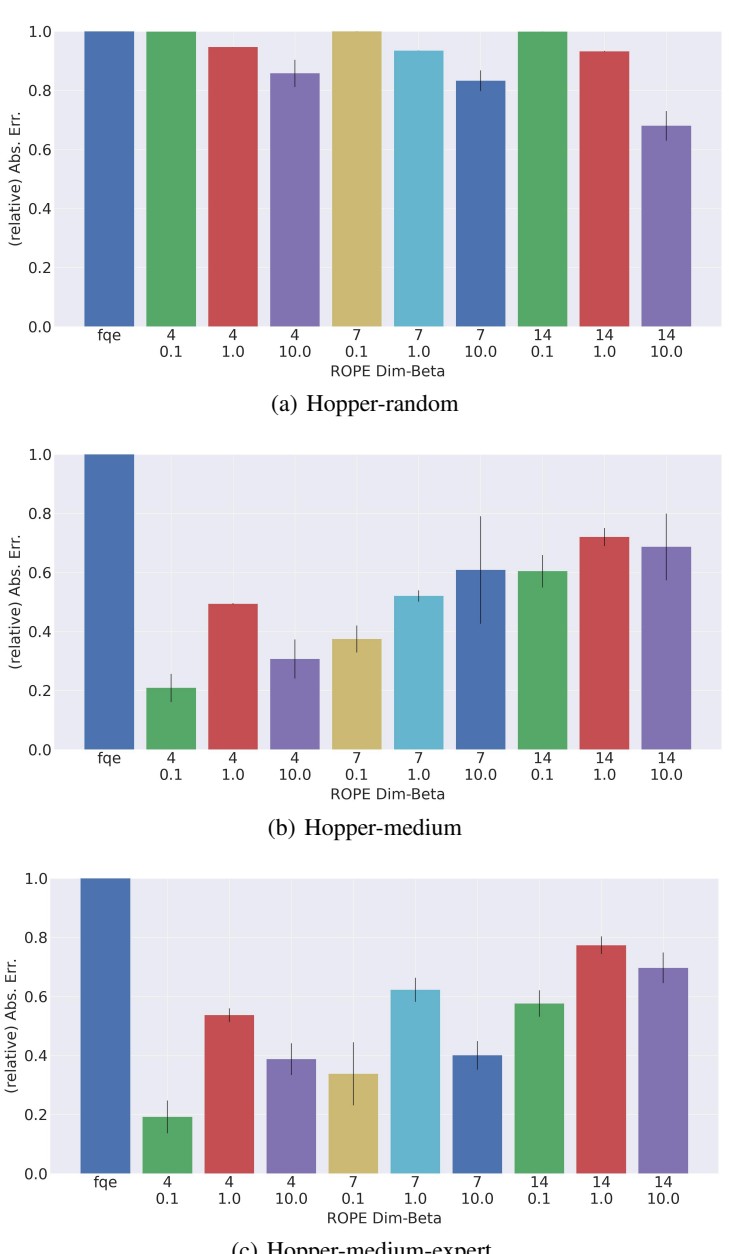

(a) Hopper-random

(b) Hopper-medium

(c) Hopper-medium-expert

Figure 10: FQE vs. ROPE when varying ROPE's encoder output dimension (top) and $\beta$ (bottom) on the D4RL datasets. IQM of errors are computed over 20 trials with 95% confidence intervals. Lower is better.

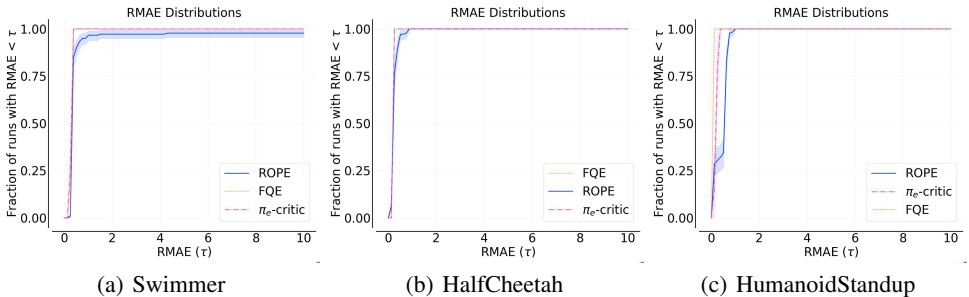

Figure 11: RMAE distributions across all runs and hyperparameters for each algorithm, resulting in $\geq 20$ runs for each algorithm. Shaded region is 95% confidence interval. Larger area under the curve is better.

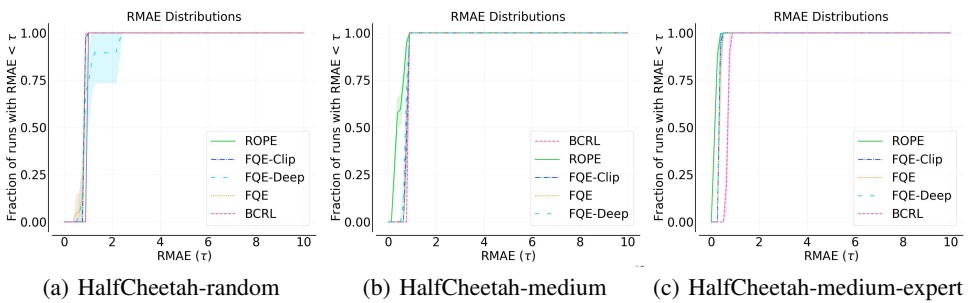

Figure 12: RMAE distributions across all runs and hyperparameters for each algorithm, resulting in $\geq 20$ runs for each algorithm. Shaded region is 95% confidence interval. Larger area under the curve is better.

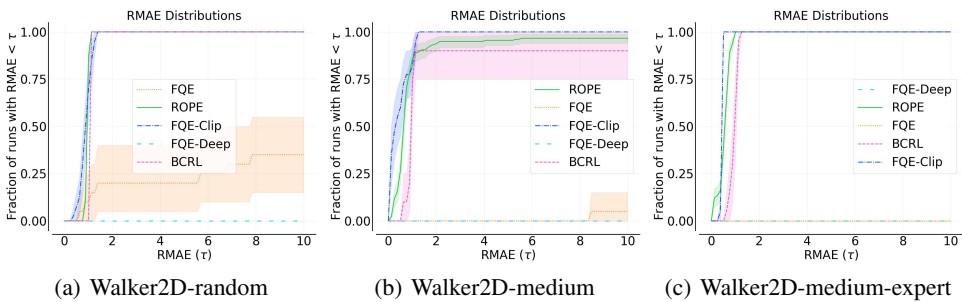

Figure 13: RMAE distributions across all runs and hyperparameters for each algorithm, resulting in $\geq 20$ runs for each algorithm. Shaded region is 95% confidence interval. Larger area under the curve is better.

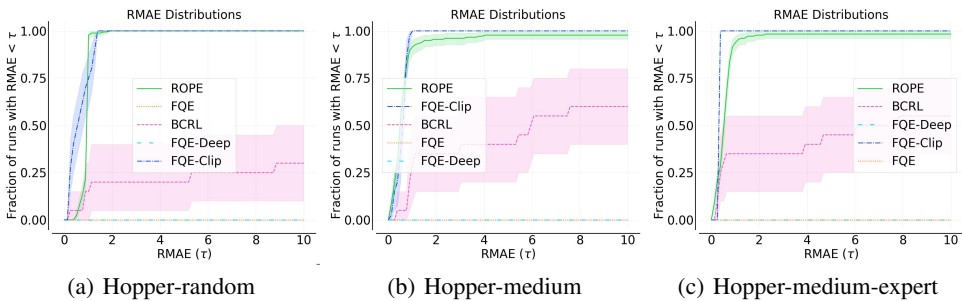

Figure 14: RMAE distributions across all runs and hyperparameters for each algorithm, resulting in $\geq 20$ runs for each algorithm. Shaded region is 95% confidence interval. Larger area under the curve is better.

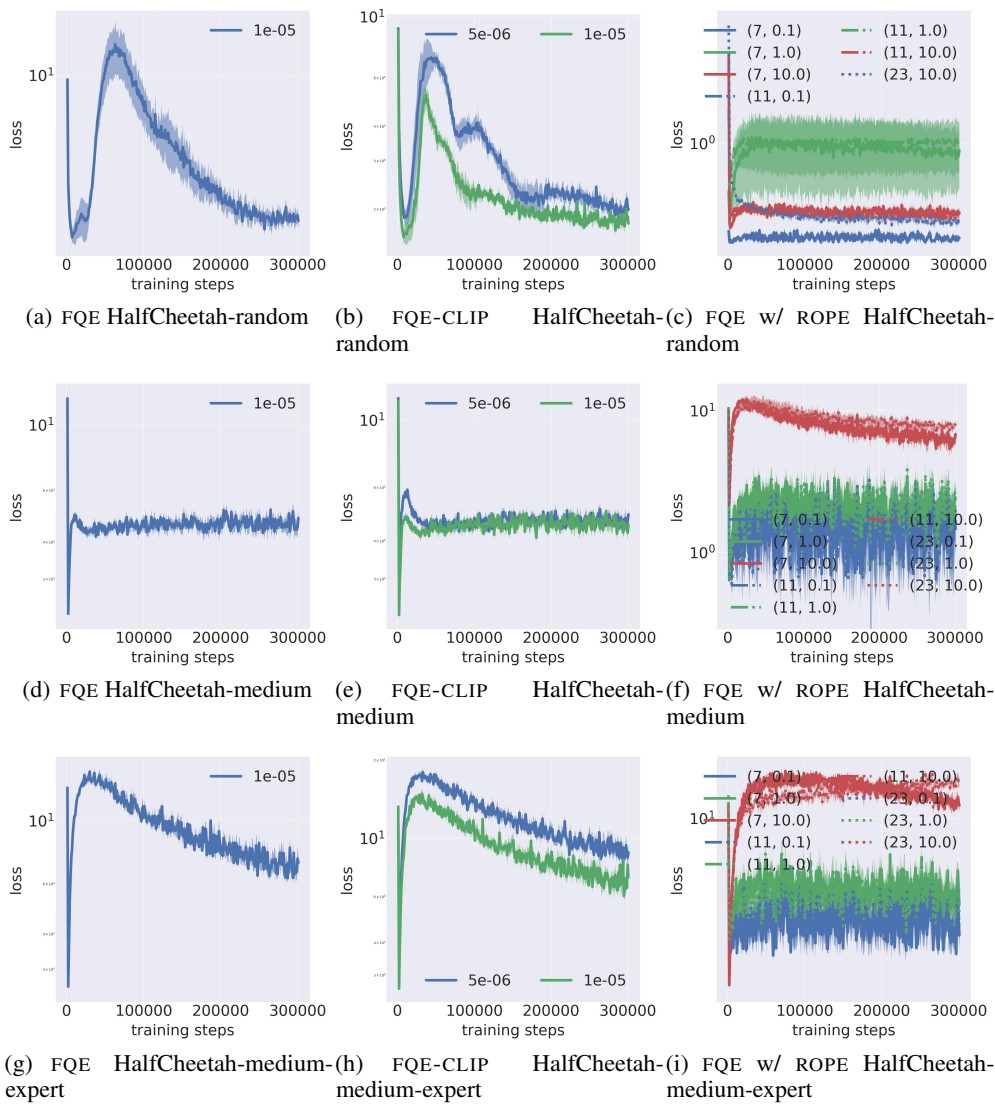

Figure 15: FQE training loss vs. training iterations on the D4RL datasets. IQM of errors for each domain were computed over 20 trials with 95% confidence intervals. Lower is better. Vertical axis is log-scaled.

### D.2.6 Understanding the ROPE Representations

In this section, we try to understand the nature of the ROPE representations. We do so by plotting the mean of the: 1) mean feature dimension and 2) standard deviation feature dimension. For example, if there $N$ state-action pairs, each with dimension $D$, we compute the mean and standard deviation feature dimension for each of the $D$ dimensions across the $N$ examples, and then compute the mean along the $D$ dimensions. If the standard deviation value is close $0$, it indicates that there may be a representation collapse. See Figure 19.

### D.3 Hardware For Experiments

For all experiments, we used the following compute infrastructure:

- Distributed cluster on HTCondor framework
- Intel(R) Xeon(R) CPU E5-2470 0 @ 2.30GHz
- RAM: 7GB

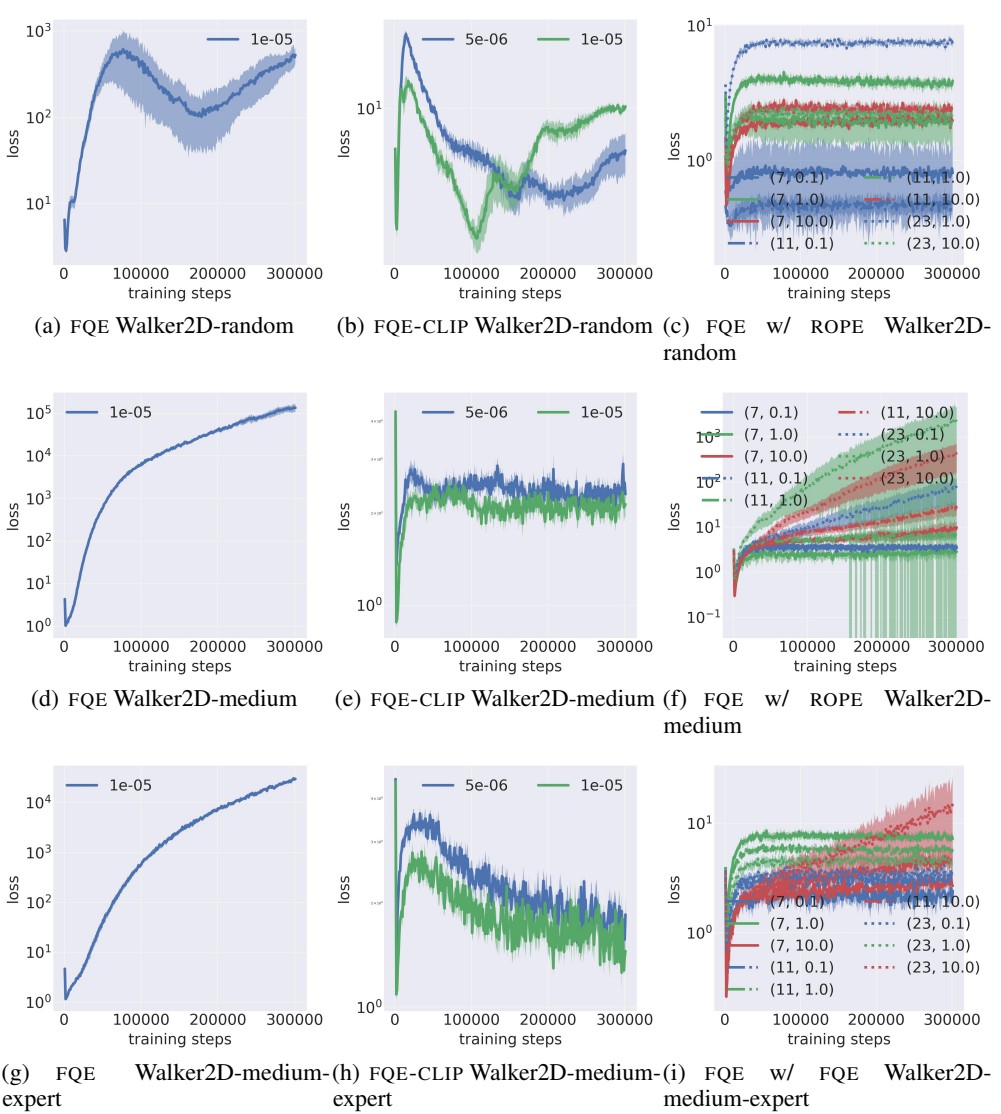

Figure 16: FQE training loss vs. training iterations on the D4RL datasets. IQM of errors for each domain were computed over 20 trials with 95% confidence intervals. Lower is better. Vertical axis is log-scaled.

- Disk space: 4GB

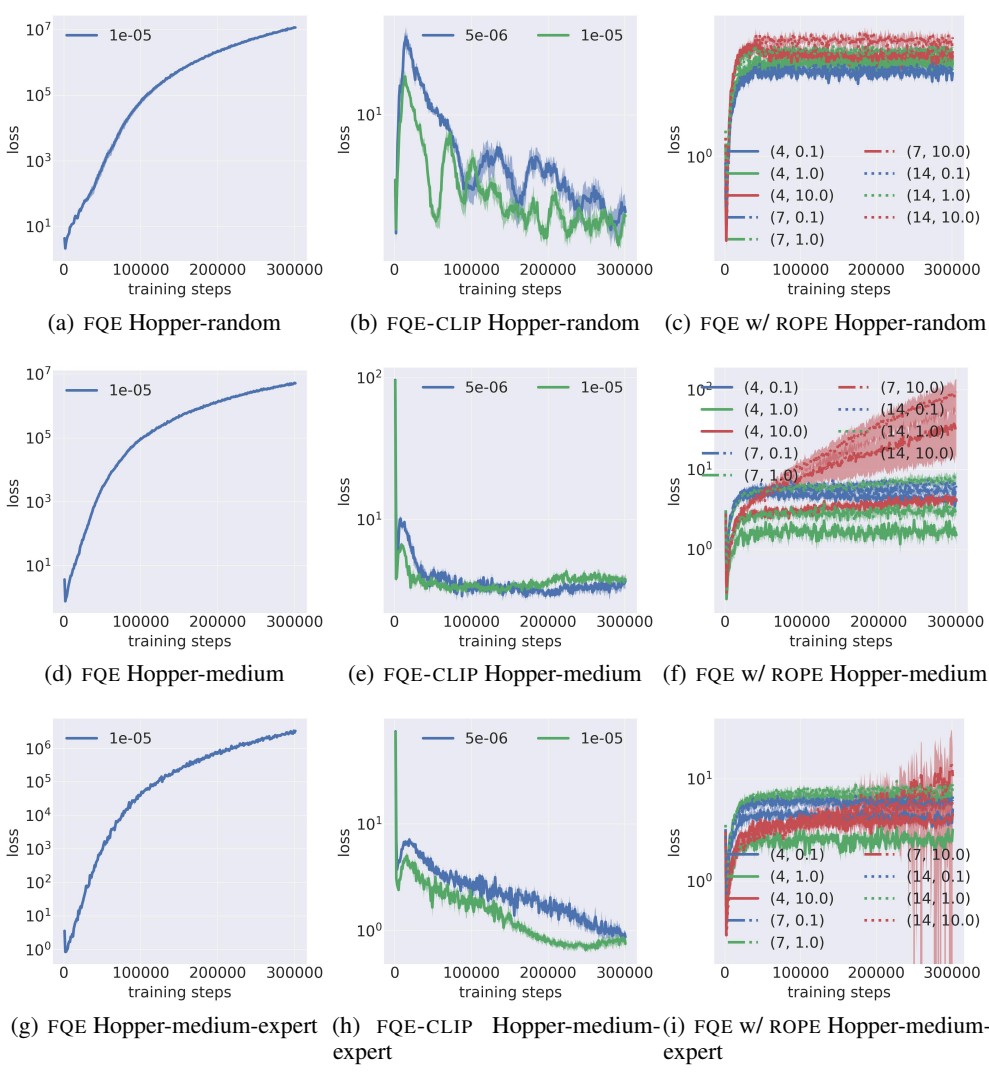

Figure 17: FQE training loss vs. training iterations on the D4RL datasets. IQM of errors for each domain were computed over 20 trials with 95% confidence intervals. Lower is better. Vertical axis is log-scaled.

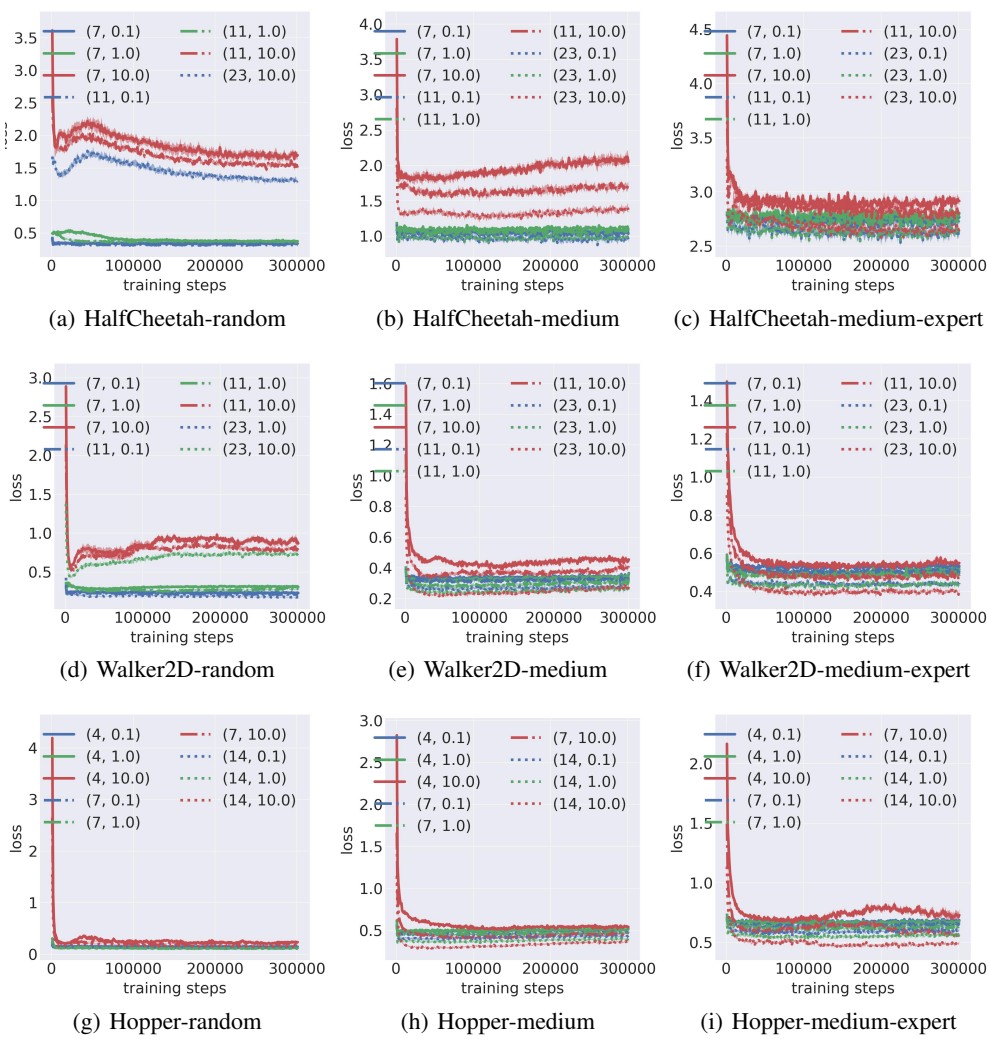

Figure 18: ROPE training loss vs. training iterations on the D4RL datasets. IQM of errors for each domain were computed over 20 trials with 95% confidence intervals. Lower is better.

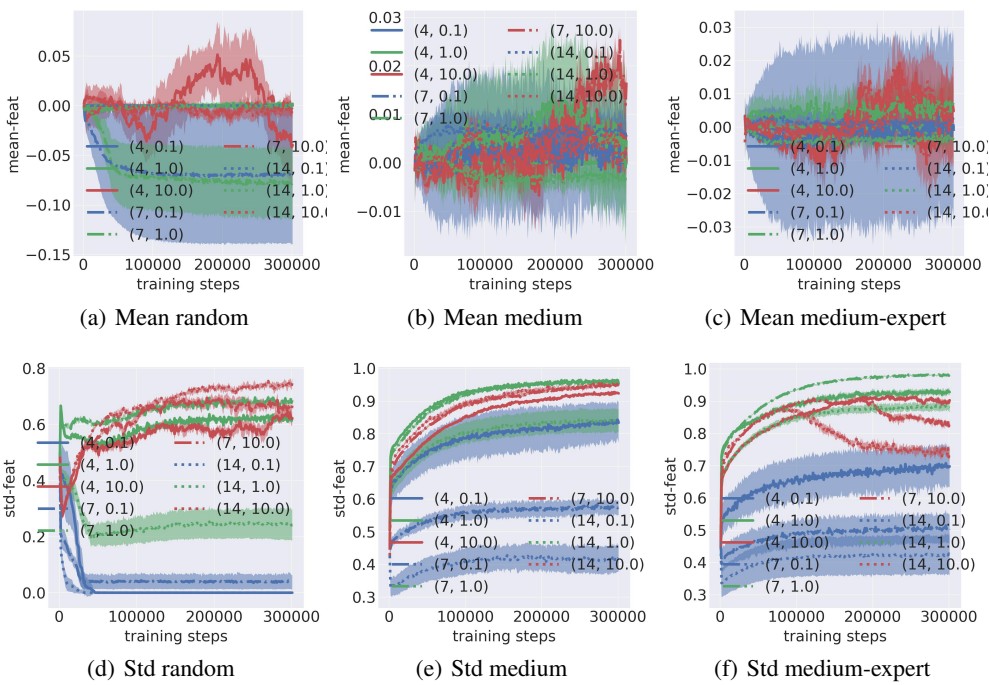

Figure 19: Mean of feature dimension stats vs. training iterations on the D4RL Hopper dataset. IQM of errors for each domain were computed over 20 trials with 95% confidence intervals.

