# OpenReview forum: "State-Action Similarity-Based Representations for Off-Policy Evaluation"
_NeurIPS.cc/2023/Conference — NeurIPS 2023 poster_

### Official Review · Reviewer_h5zX · 2023-07-05

**Soundness:** 3 good
**Presentation:** 2 fair
**Contribution:** 2 fair
**Rating:** 6
**Confidence:** 4

**Summary:**

This paper introduces a new diffuse-metric for measuring behavioral similarity between state-action pairs for OPE, named ROPE. ROPE is used to learn state-action representations using available offline data. Theoretically, this metric can bound the OPE error. Empirically, ROPE boosts the data-efficiency of FQE and achieves lower OPE error than other OPE-based representation learning algorithms. It is claimed that this work is the first that successfully uses representation learning to improve the data-efficiency of OPE.

**Strengths:**

(1) There is a certain degree of application innovation in this work. It is claimed that this work is the first that successfully uses representation learning to improve the data-efficiency of OPE.

(2) The research on related works is relatively comprehensive. The techniques involved in the proof are interesting and completely non-trivial. The maths overall seem correct and fully rigorous.

(3) The main body of the paper is well-written and easy to follow.

(4) This work enhances the data-efficiency of OPE methods through representation learning, which is of great significance for OPE methods.


**Weaknesses:**

(1) While the main body of the paper is well-written, there is space for improvement. I defer some of my issues in the appendix to "Questions".

(2) Plots can be improved by:
         Improve colour-scheme by taking into consideration colorblindness. For instance, avoid red-green-blue combination (see e.g. https://davidmathlogic.com/colorblind/#%23D81B60-%231E88E5-%23FFC107-%23004D40 for more details).

(3) There are a few misprints/suggestions in the text of the main body that I spotted:

Line 142: “s1`, s2`\~P,a1`,a2`\~pi_e”should be ” s1`~P, s2`\~P, a1`~pi_e, a2`\~pi_e “ or “s1`,s2`\~P; a1`,a2`\~pi_e”.

Line 167: “d_pi_e(s1,a1 , ;s2,a2)” should be “d_pi_e(s1,a1;s2,a2)”.


**Questions:**

(1) What is the meaning of △ in line 56? There seems to be no note.

(2) This work proposed a State-Action Behavioral Similarity Metric, of which the core is to learn the State-Action Representations. Can this State-Action Representations only be used for Off-Policy Evaluation？What is the effect of using this method to fit Q-function for general reinforcement learning algorithm.

(3) In lemma2 and theorem 2, why can the existence of the difference upper-bounds demonstrate the learned representations can help FQE estimateρ(pi_e) ?

(4) The answers of the three questions proposed in Empirical Study is not obvious. Which sections correspond to the answers?

(5) In Figure 1(c), what is the meaning of 0.7 circled in red? Is it the reference state-action (s*, a*)?


**Limitations:**

Limitations are explicitly discussed by the paper and the authors have partially addressed them. As far as I can see, has no potential negative societal impact.

---

> ### Author Rebuttal · Authors · 2023-08-09
>
> Thank you for acknowledging the merits of our work and for your suggestions. We address the concerns from the Weaknesses and Questions sections below.
>
> Weaknesses
> - Re: general comment. Thank you for the suggestions. We address them in the questions section and will make the improvements for the camera-ready version.
> - Re: plot color-scheme. Thank you very much for that useful suggestion. We will definitely update the paper based on this for the camera-ready.
> - Re: misprint. Our notation actually meant to indicate what you suggested, but we will be clearer in the camera-ready.
>
> Questions
> - Re: $\Delta$. $\Delta(X)$ means the set of all probability distributions over the set X. We will add a note in the camera-ready.
> - Re: applying ROPE to general RL. ROPE can be easily extended to the general RL setting. Since we are specifically focused on OPE, the next actions, $a_1’$ and $a_2’$, are samples from the fixed evaluation policy, $\pi_e$. To use ROPE in the general RL setting, one can sample $a_1’$ and $a_2’$ from $\pi$, which is the control policy improving over time, which is essentially a state-action version of MICO [1].
> - Re: understanding theory result. Our ultimate goal is to ensure that the learned representation satisfies realizability i.e. the representation supports estimating the true $q^{\pi_e}$. With Lemma 1 and Theorem 2, we have it that the learned action-value function after applying ROPE+FQE satisfies realizability, and the error in satisfying this is a function of the error ($\epsilon$) in the encoders ability to group state-action pairs. If the error is zero, then realizability is satisfied, which means the representation learned with ROPE estimates $\rho(\pi_e$).
> - Re: questions to experiment correspondence. We will clarify to avoid confusion in the camera-ready. To be precise, Q1 is answered by Section 4.2.1 and our global response, Q2 is answered by Section 4.2.2 (Figure 2), and Q3 is answered by Section 4.2.3 (Figure 3 and 4).
> - Re: clarification of result. The red circle indicates the reference state-action, $(s^*,a^*)$, yes. 0.7 indicates the normalized distance between $(s^*,a^*)$ and itself. Since we are dealing with diffuse metrics [1], self-distances may be non-zero as indicated in Line 248.
>
> [1] Castro P S, Kastner T, Panangaden P, et al. MICo: Improved representations via sampling-based state similarity for Markov decision processes[J]. Advances in Neural Information Processing Systems, 2021.

---

> > ### Comment · Reviewer_h5zX · 2023-08-17
> >
> > I am pleased with the clarity with which the authors addressed the questions I raised in my initial review.  The authors' responses in the rebuttal have provided lucid explanations to the concerns I had, accompanied by detailed justifications and elaborations. This has significantly bolstered the comprehensibility and scientific validity of the paper.

---

> > > ### Author Response · Authors · 2023-08-17
> > >
> > > We thank you for your response and for appreciating the merits of our work.

---

### Official Review · Reviewer_BFUR · 2023-07-06

**Soundness:** 3 good
**Presentation:** 3 good
**Contribution:** 3 good
**Rating:** 7
**Confidence:** 4

**Summary:**

This paper introduces an OPE-tailored state-action behavioral similarity metric that acts as a new loss for representation learning that can be used to learn a encoder for the state-action features in place of the original features.

**Strengths:**

-- Very well written paper

-- Interesting contribution to OPE

**Weaknesses:**

-- Little discussion about why state-representation then OPE is an easier task than OPE itself

**Questions:**

(1) It is clear that the distance metric is derived from triangle inequality on the difference in action-value functions. Can you theoretically justify why learning the representation and then plugging into FQE is an 'easier' task than just doing FQE? If it is not easier, why should we prefer it?

(2) Sometimes FQE (which i assume is the "identity" in your plots) performs competitively with (or outperforms) ROPE. How can we anticipate this? I really want the OPE community to think about robustness of algorithms.

---

> ### Author Rebuttal · Authors · 2023-08-09
>
> Thank you for appreciating our work. We address your comments below.
>
> Weaknesses:
> - Re: easiness of representation learning vs. OPE. This is a very good point and, as far as we are aware, is an open question as to when is it easier to learn a representation and plug it into FQE vs. applying FQE directly. In our paper, we do not claim that applying ROPE + FQE is easier, but we hypothesize that representation learning may help as a form of regularization where learned (s,a) representations that are similar based on the distance metric are biased towards a common solution. This is an exciting future direction that we would like to explore.
>
> Questions
> - See above (same as Weakness point).
> - Re: comparison to vanilla FQE. Thank you for this question. In the evaluated settings, we found that ROPE outperforms vanilla FQE. We suspect that vanilla FQE will perform competitively with ROPE+FQE when most state-action pairs are considered dissimilar in terms of the ROPE metric, which would lead to minimal clustering of state-action pairs, thus effectively causing ROPE+FQE to function as vanilla FQE. Further investigating this interesting direction would require understanding the intricate interplay among the environment, distribution shift, and data coverage etc.
>   - Re: robustness: We share your view about the need for robustness in OPE algorithms. We found one of the advantages of ROPE to be that it was robust to hyperparameter tuning (Section 4.2.3), which is especially important since hyperparameter tuning is difficult in the OPE setting.

---

### Official Review · Reviewer_KriT · 2023-07-07

**Soundness:** 3 good
**Presentation:** 3 good
**Contribution:** 3 good
**Rating:** 5
**Confidence:** 3

**Summary:**

The paper introduces a method to enhance the data-efficiency of the fitted q-evaluation (FQE) algorithm in off-policy evaluation (OPE) for reinforcement learning. They propose using a learned encoder and an OPE-tailored state-action behavioral similarity metric to transform the fixed dataset, improving the representation learning process. Theoretical bounds on OPE error are derived, and empirical results demonstrate the effectiveness of the proposed method in improving data-efficiency and reducing OPE error compared to other approaches.

**Strengths:**

The paper demonstrates several strengths:

1. The paper addresses an important research direction in the field of off-policy evaluation (OPE) by focusing on enhancing data-efficiency through representation learning. This contributes to the advancement of OPE methods.

2. By learning representations based on a behavioral metric, the proposed approach avoids the direct use of importance sampling, which can introduce large variance in OPE. This innovative technique improves the stability and reliability of the OPE process.


3. The paper provides theoretical analysis, demonstrating the effectiveness of the proposed algorithm. This contributes to the understanding of the underlying principles and supports the validity of the approach.

4. The paper presents numerous experimental results, validating the effectiveness of the proposed method. These empirical findings provide strong evidence of the improvements achieved in terms of data-efficiency and OPE error reduction.



**Weaknesses:**



1. My major concern is that the paper lacks a clear and intuitive explanation or discussion on why learning state-action representations improves data-efficiency in OPE. Providing a more intuitive explanation or discussing the underlying reasons for this benefit would enhance the clarity and understanding of the proposed approach.

2. It would be beneficial to include an illustration and an algorithm for the proposed paradigm in the main text. These visual aids would help readers grasp the key concepts and the implementation details more easily.

3. The paper overlooks a related work [1] that focuses on learning pseudometric-based behavioral representations for offline RL. Including a discussion of this work would enhance the completeness of the literature review and provide a more comprehensive understanding of the research landscape.

[1] Learning pseudometric-based action representations for offline reinforcement learning. ICML 2022.

**Questions:**

Considering the mentioned weaknesses, the paper raises the following questions:


 Can the authors provide a more intuitive explanation, discussion or visualizations regarding the benefits of learning state-action representations for data-efficiency in OPE? Addressing this question would enhance the clarity and understanding of the proposed approach.

Furthermore, I would like to emphasize that addressing these concerns would significantly contribute to the improvement of my evaluation and, potentially, my overall score for the paper.

**Limitations:**

limitations has been discussed by the authors.

---

> ### Author Rebuttal · Authors · 2023-08-09
>
> Thank you to the reviewer for their acknowledgement of the merits of our work, comments, and suggestions. Furthermore, thank you for providing an actionable suggestion for us to improve your evaluation of our work. Please also see our global response.
>
> Weaknesses:
> - Re: intuition behind data-efficiency. This is a great suggestion and we appreciate it for improving clarity of our algorithm. Please see the global response and the attached pdf for a visual that we hope will clarify how learning state-action representations can help OPE. We will definitely update the camera-ready with this new visual and elaborate on the intuition for ROPE.
> - Re: illustration/code of algorithm. Thank you for this suggestion. We have included pseudo-code in the attached pdf, and will definitely add it to the camera-ready version. Our method for learning the state-action encoder is similar to how MICO [1] learns a state encoder. The key differences between our work and [1] are:
>   - 1) we encode state-action pairs instead of states,
>   - 2) we train the encoder such that distances between the encoded representations of any two state-action pairs is equal to the ROPE distance between those pairs whereas Castro et al. use the MICO distance, and
>   - 3) we first train the encoder and then freeze its weights before applying it to encode all state-actions for FQE with the fixed dataset whereas Castro et al. train their state encoder as an auxiliary task while simultaneously learning optimal action-values as a function of that encoder.
> - Re: related work. Thank you for referencing this paper. We will indeed cite this work in the camera-ready. In terms of differences from our work: 1) their paper is focussed on learning action representations instead of state-action representations in offline RL and 2) their focus on offline RL is specifically for the control setting rather than the evaluation setting.
>
> Questions
> - Yes, following your actionable suggestion, we have included a global response. Please let us know if you have further questions or suggestions.
>
>
> [1] Castro P S, Kastner T, Panangaden P, et al. MICo: Improved representations via sampling-based state similarity for Markov decision processes[J]. Advances in Neural Information Processing Systems, 2021.

---

> > ### Comment · Reviewer_KriT · 2023-08-14
> >
> > Thanks for the response of authors, I will improve my score.

---

> > > ### Author Response · Authors · 2023-08-14
> > >
> > > Thank you very much to the reviewer.

---

### Official Review · Reviewer_y8hg · 2023-07-11

**Soundness:** 2 fair
**Presentation:** 2 fair
**Contribution:** 2 fair
**Rating:** 5
**Confidence:** 4

**Summary:**

Towards enhanced data-efficiency of the fitted q-evaluation (FQE) method, this work first proposes an OPE-tailored state-action behavioral similarity metric and then uses this metric and the fixed dataset to learn an encoder, which is used to transform the fixed dataset. Experiments on the OPE tasks illustrate that the proposed method improves the data-efficiency of FQE and obtains a lower OPE error compared to other OPE-based representation learning methods.

**Strengths:**

1.	The motivation of this work is clear, which is a problem worth studying under off-policy evaluation topic.
2.	Various experiments as well as relevant analyses are performed in this work, which illustrates the efficacy of the method.

**Weaknesses:**

1.	Some details of this work are not clear enough and some analysis is too superficial.
2.	Although the approach is superior to the other baseline, the advantage seems not obvious enough, except in the setting of $\mathcal{D}_{100}^{off}$.

**Questions:**

There are some questions about this paper:

1.	In Section 3.1, to learn a state-action representation, this work follows the method form [1]. I recommend that the authors give a supplementary explanation of this method. In addition, I wonder what is the difference between the proposed method, ROPE and the work [1]. In other words, why do the authors consider state-action representations? OPE-tailored behavioral similarity metric is unclear, and why is this metric strongly correlated with OPE?

2.	In Section 4.2.2, the authors consider three other OPE-based representation learning methods, I recommend that the authors introduce these methods in a more intuitive way, such as figures. The experimental results of each method should be analyzed, for example, why the target-phi-sa performs poorly. The current version is not clear.

3.	From Figure 3(b), the reward-only is significantly better than FQE and slightly worse than the rope. What are the details of the reward-only method? Does it involve learning state-action representations? This makes me more concerned about the need for learning state-action representations.

4.	This work mainly considers the FQE approach, and I am more concerned about the generalizability of the approach.

[1] Castro P S, Kastner T, Panangaden P, et al. MICo: Improved representations via sampling-based state similarity for Markov decision processes[J]. Advances in Neural Information Processing Systems, 2021, 34: 30113-30126.

**Limitations:**

This paper discusses the limitations of the proposed algorithm and future work.

---

> ### Author Rebuttal · Authors · 2023-08-09
>
> Thank you to the reviewer for their comments and suggestions. We answer concerns from the Weakness and Questions sections below. Please also see our global response.
>
> Weaknesses:
> - Re: general comment. We hope our response clarifies the details and we will make these improvements to the camera-ready.
> - Re: algorithm performance. We agree that MSE reduction between ROPE and the second-best method in each scenario is not always large. However, we wish to highlight that our method is the only method that gives consistently low absolute error across the considered scenarios. As noted by reviewer BFUR, robust estimation is an important property for OPE methods.
>
> Questions
> - Re: comparison to MICO, correlation to OPE. Thank you for the suggestion, we will expand on our description of MICO in the supplementary section. Below we attempt to clarify possible misunderstandings of ROPE, and we will include these in the camera-ready. Please also see our global response.
>   - Re: why state-action representations and OPE-tailored metric. We refer the reviewer to line 136-140 of the paper. When learning representations in the context of OPE, we need a way to account for the distribution shift between the evaluation policy and the policy that generated the data. We do state-action representations since this can be accounted for by simply sampling actions from $\pi_e$. On the other hand, MICO learns state representations, and designing an off-policy version of MICO may involve another technique to correct the distribution shift such as importance sampling, but this requires knowledge of the behavior policy that generated the data. Moreover, if multiple policies generated the data, then estimating this importance sampling ratio is even harder. Thus, ROPE is OPE-tailored as it accounts for the distribution shift and uses off-policy data to learn the representations while MICO does not account for the distribution shift and uses on-policy data.
> - Re: intuition of baselines, analysis. Thank you for this suggestion. We briefly answer your questions now and will definitely update the paper for the camera-ready.
>   - Algorithms. We note that these are not our contributions and so we simply used them as references, however, we can expand their description in the appendices. Our main goal is to show that ROPE makes FQE more data-efficient. We include these other baselines simply to benchmark performance of representation learning-for-OPE work:
>     - Identity: this is simply vanilla FQE applied to the dataset.
>     - BCRL: learns an encoder that outputs (s,a) representations that satisfy the theoretical conditions needed for convergent value function learning with linear function approximation. It then fixes the learned encoder to output (s,a) representations that are fed into Least Squares Policy Evaluation (LSPE) to learn $q^{\pi_e}$.
>     - Target-phi-sa: uses the critic of $\pi_e$ (from its training) as a fixed encoder to output fixed (s,a) representations that are fed into FQE to estimate $q^{\pi_e}$. Intuitively, the representations outputted by this encoder should have sufficient information to estimate $q^{\pi_e}$.
>   - Analysis of target-phi-sa. Please see lines 285-288. We simply show the performance of target-phi-sa for benchmarking purposes. However, it is known that target-phi-sa may perform poorly, which, as the authors of [3] note, is interesting because target-phi-sa contains sufficient information to perfectly represent the value of the evaluation policy and satisfy realizability.
>   - If the reviewer has further analysis that they think would strengthen the paper, we would be happy to conduct it and add it to the paper for the camera-ready.
> - Re: reward-only. For details of the reward-only method, we refer the reviewer to line 312 and lines 316-319. In short, the reward-only method is a variation of ROPE that learns state-action features but where the long-term distance component (see Eqn 1, line 147) of the ROPE metric is removed. One of the main messages (lines 316-319) of this experiment is that learning appropriate state-action representations is nuanced: in some cases it appears grouping state-actions based only on reward is sufficient to do well. In future work, we will plan to understand the intricate nature between state-action representations, distribution shift, and the nature of the environment to better understand under what conditions certain distance metrics are better than others.
> - Re: beyond FQE. Representation learning for OPE beyond FQE is a very interesting direction we hope to explore in future work. Given that FQE is one of the most successful algorithms in OPE [1, 2] and that it is a core component of many RL algorithms, we focused our first efforts specifically in trying to understand the representation learning considerations for FQE.
>
> [1] Benchmarks for Deep Off-Policy Evaluation. Fu et al. 2021.
>
> [2] Empirical Study of Off-Policy Policy Evaluation for Reinforcement Learning. Voloshin et al. 2019.
>
> [3] Instabilities of Offline RL with Pre-Trained Neural Representation. Wang et al. 2021.

---

> > ### Author Response · Authors · 2023-08-17
> >
> > Hello reviewer y8hg, we just wanted to follow-up to see if our response (and global response) clarified your concerns before the discussion period ends. If you had any further questions, we would be more than happy to answer them. Thank you again for your feedback.

---

### Author Rebuttal · Authors · 2023-08-09

Thank you to the reviewers for kind words regarding the merits of our work and helpful suggestions. Before responding to individual questions and comments, we wanted to briefly elaborate on the intuition for ROPE increasing data efficiency and how ROPE is tailored to the OPE setting. We also include figures and pseudo-code that will help clarify these points, which we will include in the camera-ready.

To build intuition, we follow prior work [1, 2, 3, 4] and make the analogy between state-action aggregation and state-action representations (representation learning can be viewed as a soft form of state-action-aggregation according to some distance metric). Under this view, ROPE is a method for grouping different state-action pairs based on similarity under the ROPE metric. If all state-action pairs in a given group have small pairwise ROPE distance then they have similar state-action representations and they are behaviorally similar (i.e, have similar rewards and lead to similar future states when following the evaluation policy) and consequently will have a similar action-value. Thus, data samples from any member of the group can be generalized for more data-efficient learning to learn the group’s shared action-value as opposed to learning the action-value for each state-action pair individually. ROPE is a metric that quantifies this notion of behavioral similarity and enables identifying when we should be able to generalize across different state-action pairs.

ROPE is OPE-tailored in that it can be learned with off-policy data as it is designed to account for the distribution shift between the fixed dataset and $\pi_e$. Prior work in learning representations based on behavioral similarity (e.g., MICO) would not suffice here as they focus on on-policy learning.

In the attached pdf, we illustrate the state-action aggregation interpretation of ROPE using the gridworld domain so as to show how the ROPE metric is suitable for learning $q^{\pi_e}$ while other reasonable metric choices are not. The attached figures show 1) the action-values of the evaluation policy, 2) how ROPE groups state-action pairs, and 3) how alternative metric choices group state-action pairs. We expect the color-coding of the action-values of $\pi_e$ to match with that of the group clusters by each metric. We highlight that only ROPE correctly groups state-action pairs together only when they share the same action-value and this enables learning $q^{\pi_e}$ as required by FQE. Other metrics group state-action pairs together with different action-values and hence would produce biased FQE estimates.

We also include the pseudo-code of ROPE + FQE in the attached pdf, which we will include in the camera-ready.

[1] Scalable methods for computing state similarity in deterministic Markov Decision Processes. Castro. 2020.

[2] Learning Invariant Representations for Reinforcement Learning without Reconstruction. Zhang et al. 2021.

[3] MICo: Improved representations via sampling-based state similarity for Markov decision processes. Castro et al. 2021.

[4] Learning pseudometric-based action representations for offline reinforcement learning. Gu et al. 2022.

---

### Decision · Program_Chairs · 2023-09-21

**Decision:**

Accept (poster)

**Comment:**

The paper introduces a method to enhance the data-efficiency of the fitted q-evaluation (FQE) algorithm in off-policy evaluation (OPE) for reinforcement learning. Paper addresses an important research question in the OPE problem of RL to enhance data-efficiency through representation learning. The paper provides theoretical analysis, demonstrating the effectiveness of the proposed algorithm. Empirical results provide good evidence of the improved data efficiency of the proposed OPE method.